# Application of a Reduced-Dimensional Model for Fluid Flow between Stacks of Parallel Plates with Complex Surface Topography

Yupeng Sun [1,2], Hafiz Muhammad Adeel Hassan [2] and Joe Alexandersen [2,*]

1    Key Laboratory of Traffic Safety on Track of Ministry of Education, Central South University, Changsha 410017, China; yupeng_sun@csu.edu.cn
2    Department of Mechanical and Electrical Engineering, University of Southern Denmark, 5230 Odense, Denmark; m.adeel16@gmail.com
*    Correspondence: joal@sdu.dk

**Abstract:** Stacked plate heat exchangers are widely used in thermal energy storage systems and a comprehensive and accurate analysis is necessary for their application and optimization. The fluid flow distribution between the plates is important to ensure even and full usage of the thermal energy storage potential. However, due to the complex topography of the plate surface, it would be computationally expensive to simulate the flow distribution in the multiple channels using a full three-dimensional model, so this work applies a reduced-dimensional model to significantly reduce the computational cost of the simulation and provides a comprehensive analysis of the effect of the internal structure on the internal flow distribution. The work extends a previously presented model to consider transient flow and a multichannel height distribution strategy to allow for simulating multiple channels between stacks of plates. Based on fully-developed flow assumptions, the three-dimensional model is reduced to a planar model, thus obtaining simulation results with satisfactory accuracy at a significantly lower computational cost. The model is verified by a three-dimensional simulation of a sliced two-channel model representing the considered system. The reduced-dimensional model gives similar results to the three-dimensional model for different geometrical and physical parameters. Lastly, the extended reduced-dimensional model is used to simulate the flow of a full two-channel model and the influence of the plate topography on the internal flow distribution is investigated through a comprehensive parametric analysis. The analysis shows that the complex topography of the plate surface eliminates the variation in inlet velocity and significantly changes the internal fluid flow, eventually resulting in a consistent velocity distribution.

**Keywords:** simplified flow model; fluid flow in gaps; complex surface topography; multichannel stack model; thermal energy storage

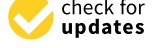



## 1. Introduction

### 1.1. Motivation

The presented work stems from a project to design a thermal energy storage (TES) system based on stacks of Compact Storage Module (CSM) plates filled with phase change material (PCM) [1,2] to be integrated into building ventilation systems [3]. TES systems play a critical role in sustainable energy production and consumption. The demand for TES has been growing in recent years due to the increasing demand for renewable energy sources and the need to reduce carbon emissions. Plate heat exchangers are commonly used in TES systems to transfer heat between solids and fluids. Plate heat exchangers consist of multiple layers of plates with undulating surfaces stacked in parallel. The topography of the plates significantly affects the fluid flow and heat transfer [4–6], and a detailed study of the topography effect is necessary for efficient improvement and design of TES systems. Simulating the fluid flow and heat transfer in plate heat exchangers is a challenging and

computationally expensive task due to the large dimensional differences between the planar dimensions of the plates and the channel heights. The high cost of simulation makes it difficult to conduct a detailed and comprehensive study of the topography effect on fluid flow in plate heat exchangers, hindering the improvement and design of TES systems. Therefore, there is a pressing need for an efficient and cost-effective method to simulate plate heat exchangers while maintaining accuracy.

### 1.2. Literature

Plate heat exchangers have been extensively studied, and numerous numerical and experimental methods have been developed to investigate their fluid flow and heat transfer characteristics [7–9]. To overcome the computational costs associated with multiple channels between stacks of plates with complex topography, various simplified methods have been used.

Mathematical modeling based on the characteristic configuration of the entire heat exchange system is a common simplification method. For example, Gut and Pinto [10] developed a mathematical model for the steady-state simulation of a gasketed plate heat exchanger with a generalized configuration, and subsequently optimized the configuration parameters [11]. Similarly, Michel and Kugi [12] presented a low-dimensional mathematical model for a plate heat exchanger that was validated experimentally. Although these models can provide useful insights into the flow characteristics of the whole system, they have difficulty capturing the local flow behavior within the system.

Projecting the system in a certain direction and simulating it in a low-dimensional model is another simplification method. For example, Zeinelabdein et al. [13] simplified the topography of the plate by projecting the three-dimensional TES model onto a two-dimensional plane and performing transient CFD simulations. While this approach obtains the overall flow characteristics of the system or the flow characteristics in a particular direction at a lower computational cost, it requires a good understanding of the system and the problem under study to ensure that the effect of the local characteristics can be neglected.

The analysis of simplified local structures based on the symmetry or repetition of the structure is also a common simplification method. For example, Tong et al. [14] selected a sliced structure in a pillow plate heat exchanger for simulation and optimized the combination parameters of the pillow plate based on the simulation results. While this method captures the effect of local structural features on the overall system, it requires a good periodic distribution of the structure, which makes it difficult to provide accurate simulation results for randomly distributed structures.

Recently, pseudo-3D methods have been used for the simulation and optimization of finned heat sinks [15,16]. The methods are based on assumptions of fully developed flow and reduce the three-dimensional flow equations to two dimensions to simplify the simulation of parallel flow. Earlier pseudo-3D methods did not properly consider out-of-plane resistance when introducing velocity absorption terms to model in-plane flow resistance. Yan et al. [17] reviewed the work presented by Borrvall and Petersson [18] and used the velocity absorption term proposed by them to represent the out-of-plane shear forces on the fluid in a two-layer model including heat transfer. Subsequently, Zhao et al. [19] proposed a three-layer model for forced convective heat transfer in a planar cooling channel, considering the out-of-plane flow resistance. It was later expanded to include turbulent forced convection heat transfer [20]. Although the pseudo-3D methods of the above studies have good accuracy, these studies assume that the fluid regions remain at the same height out-of-plane. For plate heat exchangers with complex surface topography and varying fluid region heights, the above pseudo-3D methods are unable to obtain accurate velocity distributions.

To address the above issues, Alexandersen [21] proposed a reduced-dimensional model for fluid flow between parallel plates with varying heights, considering the height variation of the fluid area. It builds on the work of Borrvall and Petersson [18] and

Gersborg-Hansen et al. [22] and modifies the mass conservation equation of the fluid by considering the height variation of the control volume so that the height variation of the fluid is considered. Through comparisons in two and three dimensions, it was demonstrated by Alexandersen [21] that the velocity distributions obtained using the traditional reduced-dimensional model (used in the previous pseudo-3D approach) differ significantly from those obtained with the full three-dimensional model when the out-of-plane height varies. In contrast, the model proposed by Alexandersen [21] considers the effects of height variation and can obtain consistent velocity distributions with the full three-dimensional model.

*1.3. Contributions*

This work proposes an extension of the reduced-dimensional model proposed by Alexandersen [21] to transient fluid flow and a multichannel height distribution strategy for the simulation of fluid flow within plate stacks of a heat exchanger.

A graphical overview of the paper is given in Figure 1. This paper combines the proposed multichannel height distribution strategy with an extended reduced-dimensional model to provide an efficient and relatively accurate approach to modeling fluid flow in plate heat exchangers in TES systems that can be applied to industrial problems.

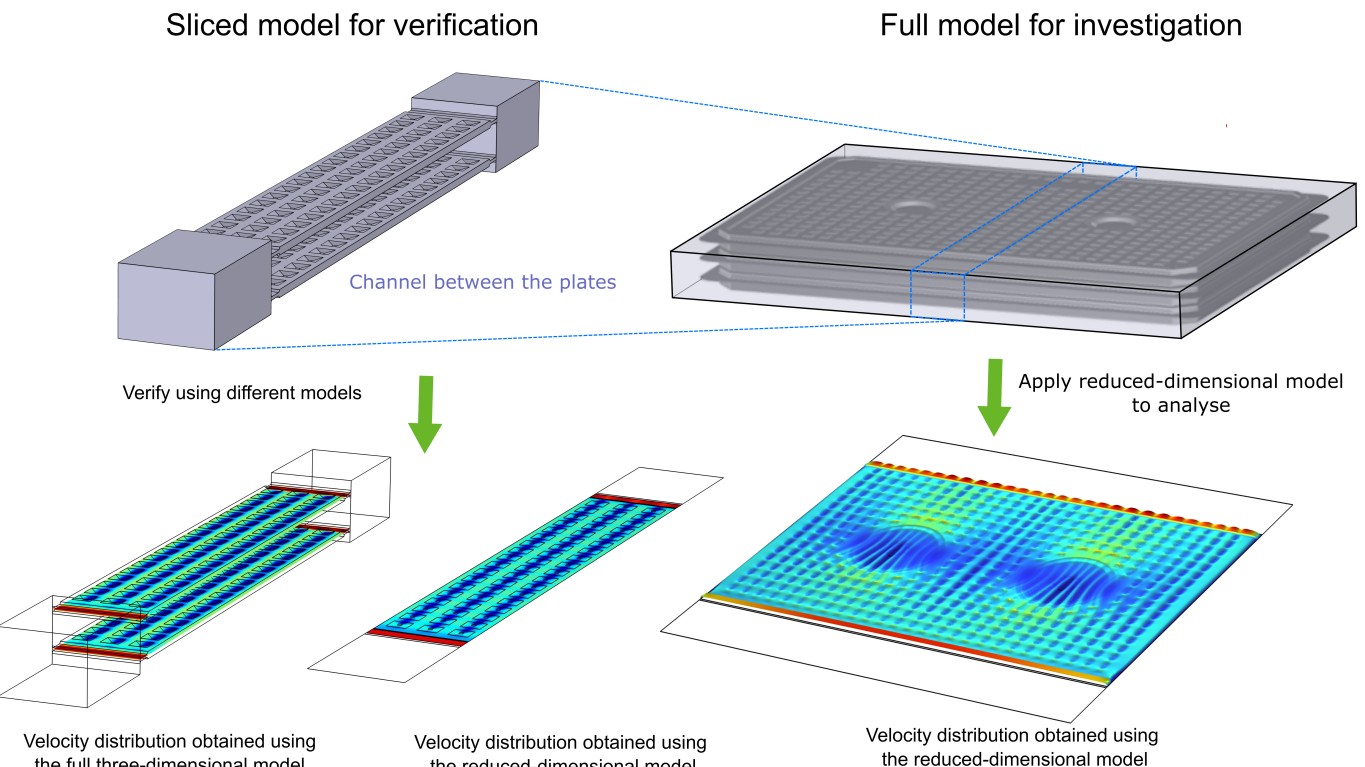

**Figure 1.** Graphical overview of the application of the reduced-dimensional model to simulate the flow between the stacked heat exchanger plates.

This work uses the presented model to perform a comprehensive parametric analysis of the effect of plate topography on internal flow in plate heat exchangers. The study focuses on a detailed analysis of velocity profiles, magnitudes, and plate dimensions. The result of the parametric analysis contributes to a deeper understanding of the complex fluid flow phenomena in plate heat exchangers and provides valuable insights into the design of thermal energy storage modules.

### 1.4. Paper Layout

The rest of the paper is organized as follows: Section 2 provides a detailed description of the sliced two-channel model and the full two-channel model studied, as well as the extended reduced-dimensional model and the multichannel height distribution strategy; Section 3 shows a comparison of velocity and pressure for the sliced two-channel model obtained using the full three-dimensional model and the reduced-dimensional model. The results of the parametric analysis of the full two-channel model using the reduced-dimensional model are also presented to investigate the effect of plate topography on fluid flow; Section 4 summarizes the findings and discusses the implications of the work for the design of TES systems.

## 2. Models and Methods

### 2.1. Thermal Storage Module

This work simulates the fluid flow within a TES system, as shown in Figure 2, and investigates the effect of the structure of the internal plates on the fluid flow. The TES system consists of two parallel stacks of heat exchange plates filled with phase change material: each stack contains 24 plates with a 1.5 mm channel between the plates for fluid flow. The planar dimensions of the plates are much larger than the channel height, and numerous raised dimples exist to provide stiffening to carry the internal phase change material and increase the contact area with the fluid.

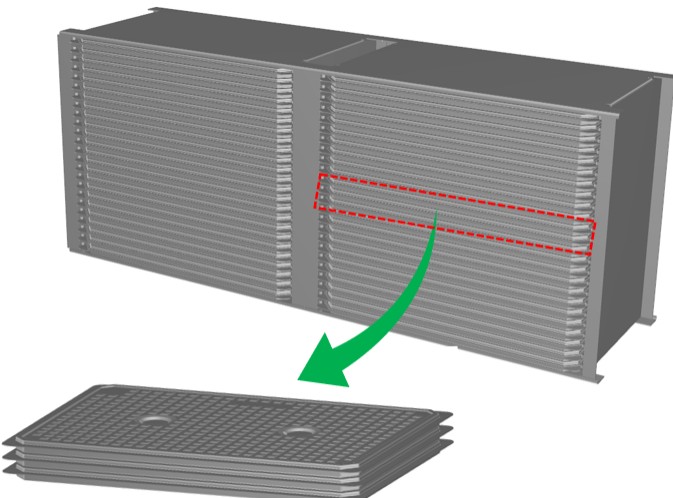

**Figure 2.** Adaptation of full two-channel model from TES module consisting of stacks of plates.

Using a full three-dimensional model for the simulation of the above system is not a feasible approach. Accurate simulation of the effect of raised dimples on the flow requires the mesh to be sufficiently fine locally, and the large dimensional span between the plane dimensions and the channel height will cause the number of elements to be extremely large. This means that it is highly impractical to consider the entire plate stack using a full three-dimensional model without consuming significant computational resources. Therefore, a simplification of the model is required. This work will focus on a two-channel model between three plates (as illustrated in Figure 2) and a sliced version will be analyzed using a three-dimensional model and a reduced-dimensional model. After verifying the accuracy of the reduced model, the full two-channel model will be analyzed using only the reduced-dimensional model.

### 2.2. Sliced Three-Dimensional Model

Due to the symmetry and approximate periodicity of the plate topographies, a sliced two-channel model, as shown in Figure 3, is used to reduce computational costs for verification analyses. The dimensions and parameters are given in Table 1. The model consists of two fluid channels sandwiched between three plates. There is an inlet area and an outlet area at the front and back of the channels, respectively. The extended inlet domain is needed to capture the flow distribution from the inlet region into the multiple channels between the plates, while a longer outlet area ensures the mitigation of reverse flow, as well as providing visual information about vortex generation and shedding as the air jets come out.

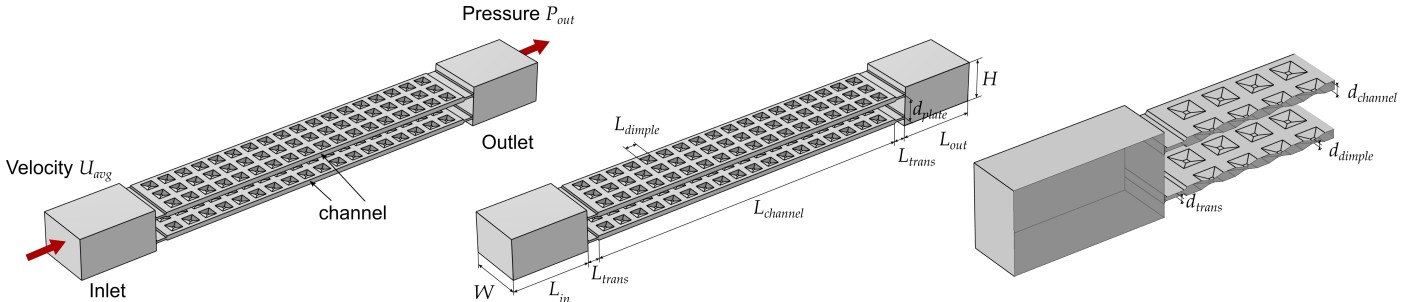

**Figure 3.** Boundary conditions and dimensions for a three-dimensional sliced two-channel model. Plates are not modeled.

**Table 1.** Dimensions of the sliced two-channel model (see Figure 3) and the physical parameters of the incoming air at 27 °C [2].

| Parameters | Symbols | Values | Units |
|---|---|---|---|
| Width | $W$ | 43 | mm |
| Height | $H$ | 32 | mm |
| Length of inlet | $L_{in}$ | 60 | mm |
| Length of outlet | $L_{out}$ | 60 | mm |
| Length of channels | $L_{channel}$ | 258 | mm |
| Length of transitions | $L_{trans}$ | 10 | mm |
| Length of dimples | $L_{dimple}$ | 9 | mm |
| Thickness of channels | $d_{channel}$ | 3.5 | mm |
| Thickness of dimples | $d_{dimple}$ | 1.5 | mm |
| Thickness of transitions | $d_{trans}$ | 1.5 | mm |
| Thickness of plates | $d_{plate}$ | 15 | mm |
| Density of fluid | $\varrho$ | 1.177 | kg/m$^3$ |
| Viscosity of fluid | $\mu$ | $1.846 \times 10^{-5}$ | kg/(m · s) |
| Average inlet velocity | $U_{avg}$ | 0.05 | m/s |

The fluid flow within this three-dimensional model is governed by the incompressible, steady-state Navier–Stokes equations, which are established as follows:

$$\varrho u_j \frac{\partial u_i}{\partial x_j} - \mu \frac{\partial}{\partial x_j} \left( \frac{\partial u_i}{\partial x_j} + \frac{\partial u_j}{\partial x_i} \right) + \frac{\partial p}{\partial x_i} = 0 \tag{1a}$$

$$\varrho \frac{\partial u_i}{\partial x_i} = 0 \tag{1b}$$

where $i, j \in \{1, 2, 3\}$, $u_i$ is the velocity component in the $x_i$ direction, $\varrho$ is the density of the fluid, $\mu$ is the dynamic viscosity of the fluid, and $p$ is the fluid pressure.

### 2.3. Reduced Two-Dimensional Approximation

Although the sliced two-channel model reduces the computational costs by reducing the size of the computational domain, there are limitations. When the overall model is asymmetric or when the performance of the overall structure needs to be studied, the sliced model cannot satisfy the needs of the study. This work, therefore, uses a reduced-dimensional model to reduce the three-dimensional problem to a planar problem. The reduced-dimensional model provides computational savings, which allows the study of the fluid flow in the full channels and the study of asymmetric flow.

### 2.3.1. Reduced-Dimensional Model

This paper applies the reduced-dimensional model proposed by Alexandersen [21] with the treatment of channel heights to the problem of flow between multichannel stacks of plates. The model is based on the assumption of fully-developed planar flow and represents three-dimensional velocity fields using a planar velocity and a constant out-of-plane profile. When substituted into the three-dimensional governing equations, this reduces the dimension of the model by one. The model considers the height variation of the control volume and modifies the mass conservation equation, which allows the model to be applied to the simplification of flow problems with varying channel heights:

$$\bar{\varrho}\bar{u}_j \frac{\partial \bar{u}_i}{\partial x_j} - \mu \frac{\partial}{\partial x_j}\left(\frac{\partial \bar{u}_i}{\partial x_j} + \frac{\partial \bar{u}_j}{\partial x_i}\right) + \frac{\partial \bar{p}}{\partial x_i} - \frac{10\mu}{h(\mathbf{x})^2}\bar{u}_i = 0 \tag{2a}$$

$$\tilde{h}(\mathbf{x})\frac{\partial \bar{u}_i}{\partial x_i} + \bar{u}_i \frac{\partial \tilde{h}(\mathbf{x})}{\partial x_i} = 0 \tag{2b}$$

where $i, j \in \{1, 2\}$, $\bar{\varrho} = \frac{6}{7}\varrho$ is the effective density, $h(\mathbf{x})$ is the spatially-varying out-of-plane height of a single channel, $\tilde{h}(\mathbf{x})$ is the spatially-varying total height of the channels in the stack, $\bar{u}_i$ are the in-plane velocity components, $\bar{p} = \frac{5}{4}p$ is the scaled pressure field, and the last term of Equation (2a) is the out-of-plane viscous resistance whose magnitude varies with the local channel height. An important difference between the full three-dimensional model and the plane reduced-dimensional model is that the out-of-plane height enters as a continuous field in the latter, not as a physical out-of-plane dimension as in the former.

The novelty in the herein-proposed multichannel height distribution strategy is the use of different height functions in the momentum and mass conservation equations, Equations (2a) and (2b), respectively. For the momentum conservation equations, only the local height of the channel is important to include the out-of-plane viscous resistance. However, for the mass conservation equation, the total fluid volume must be used, which consists of the sum of the heights of all channel between the parallel plates:

$$\tilde{h}(x_1, x_2) = m \cdot h(x_1, x_2) \tag{3}$$

where $h$ is the local height of a single-layer channel, $m$ is the number of channels, and $\tilde{h}$ is the height used for the calculation of mass conservation in the reduced-dimensional model. The single channel height is used in the momentum equation because the magnitude of out-of-plane resistance only depends on the local height of the single channel. On the other hand, the total sum of heights of all channels is used in the mass conservation equation to satisfy the conservation of mass flow in the distribution among multiple channels.

### 2.3.2. Two-Dimensional Approximation for Sliced Two-Channel Model

To verify the effectiveness of the adopted reduced-dimensional model and the multi-channel height distribution strategy, the sliced two-channel model is reduced in dimension and compared with the previous three-dimensional results. The geometry of the reduced-dimensional model of the sliced three-plate model is shown in Figure 4. The dimensional parameters of the features are all consistent with the three-dimensional model given in Table 1.

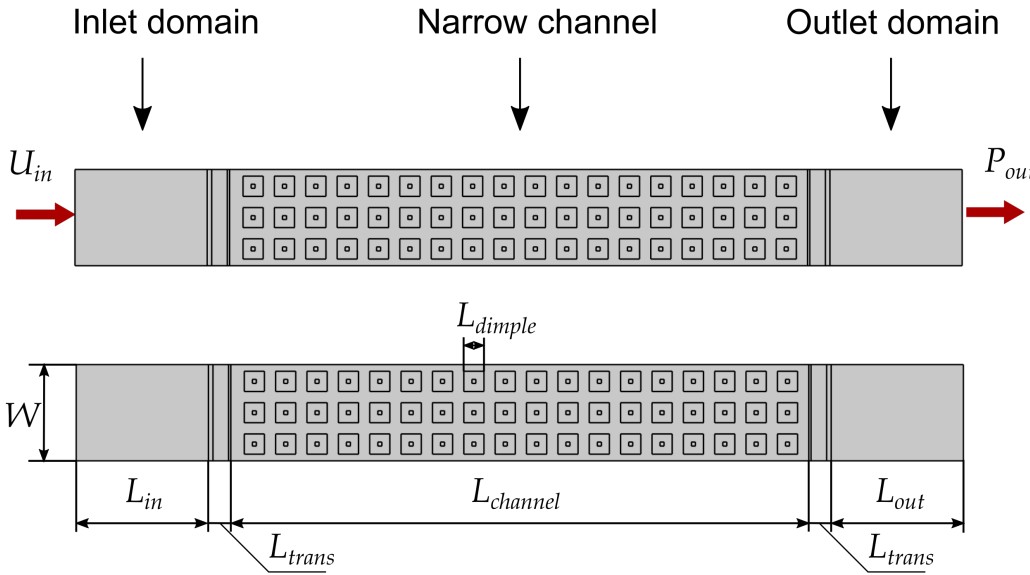

**Figure 4.** Boundary conditions and dimensions for the sliced two-channel model for the reduced-dimensional model.

The heights of the sliced two-channel model are smoothed using the following PDE filter [23] to decrease the error introduced by height variations [21] on the reduced-dimensional model:

$$-r^2 \nabla^2 h + h = \bar{h} \tag{4}$$

where *r* is the filter factor, set to 3 mm for the sliced two-channel model and 5 mm for the full two-channel model. $\bar{h}$ is the height distribution defined by the features and *h* is the continuous height field obtained after the filter. According to the proposed multichannel height distribution strategy, the height distribution of the reduced-dimensional model is shown in Figure 5. Figure 5a shows the heights for the momentum equations, $h(x_1, x_2)$, and Figure 5b shows the heights for the mass conservation equation, $\tilde{h}(x_1, x_2)$.

### 2.3.3. Two-Dimensional Approximation for Full Two-Channel Model

This paper uses a reduced-dimensional model to study the flow of air across and between the plates. As can be seen in Figures 2 and 6, the plates have numerous dimples and there are two circular depressions in the middle of the plates. To ensure that the simulation results are accurate, the reduced-dimensional model is modeled with the same structural details as the three-dimensional model, but in a planar fashion with a height field defined by the features and the filter. The dimensional parameters of the model and the properties of the fluid are shown in Table 2.

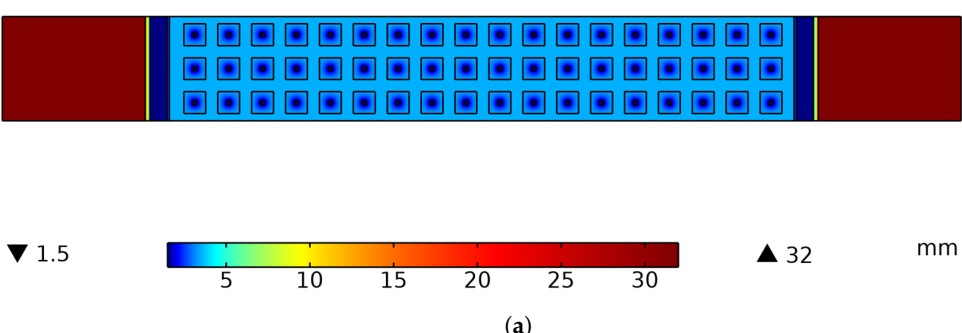

▼ 1.5          5   10   15   20   25   30          ▲ 32     mm

(**a**)

**Figure 5.** *Cont.*

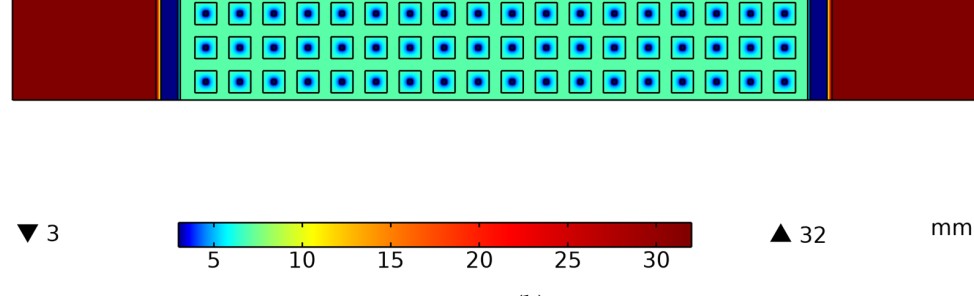

**(b)**

**Figure 5.** Height distribution of sliced two-channel model for reduced-dimensional model. (**a**) Height for momentum equations, $h(x_1, x_2)$; (**b**) height for mass conservation, $\tilde{h}(x_1, x_2)$.

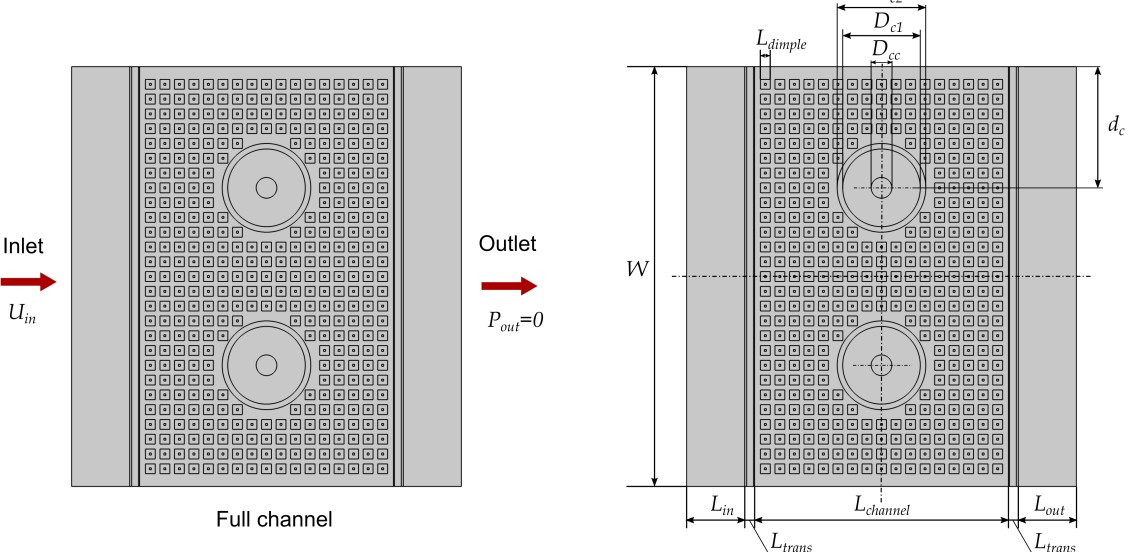

**Figure 6.** Boundary conditions and dimensions for the full two-channel model for reduced-dimensional model.

**Table 2.** Dimensions of the full two-channel model and the physical parameters of air at 27 °C [2].

| Parameters | Symbols | Values | Units |
|---|---|---|---|
| Width | $W$ | 426 | mm |
| Height | $H$ | 32 | mm |
| Length of inlet | $L_{in}$ | 60 | mm |
| Length of outlet | $L_{out}$ | 60 | mm |
| Length of channels | $L_{channel}$ | 258 | mm |
| Length of transitions | $L_{trans}$ | 10 | mm |
| Length of dimples | $L_{dimple}$ | 9 | mm |
| Thickness of channels | $d_{channel}$ | 3.5 | mm |
| Thickness of dimples | $d_{dimple}$ | 1.5 | mm |
| Thickness of transitions | $d_{trans}$ | 1.5 | mm |
| Thickness of plates | $d_{plate}$ | 15 | mm |
| Diameter of small circle | $D_{cc}$ | 21 | mm |
| Diameter of circle 1 | $D_{c1}$ | 78.8 | mm |
| Diameter of circle 2 | $D_{c2}$ | 90 | mm |
| Distance between circle center and edge | $d_c$ | 123 | mm |
| Density of fluid | $\varrho$ | 1.177 | kg/m$^3$ |
| Viscosity of fluid | $\mu$ | $1.846 \times 10^{-5}$ | kg/(m · s) |
| Average inlet velocity | $U_{avg}$ | 0.189 | m/s |

The rectangular areas at the left and right ends of the model are the inlet and outlet, both having heights of the two channels plus the thickness of one and two half plates. The depths of the dimples, the channel heights, and the depths of the circular area are all obtained from measurements of a physical plate. The heights of the area between the dimples and the heights between the circular depression and the plate are smoothed with the PDE filter, as shown in Equation (4). According to the proposed height assignment strategy, the height distribution of the reduced-dimensional model is shown in Figure 7. Figure 7a shows the height for the momentum equations, and Figure 7b shows the height used for the mass conservation equation.

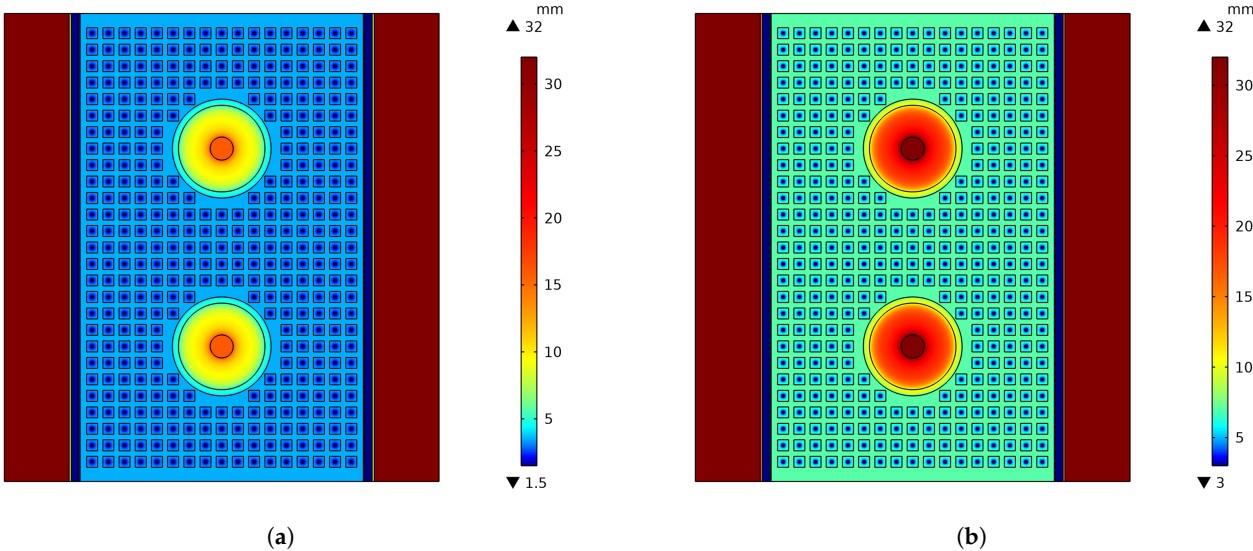

(**a**)                                                                                    (**b**)

**Figure 7.** Height distributions of full two-channel model for reduce-dimensional model. (**a**) Heights for momentum equation, $h(x_1, x_2)$; (**b**) heights for mass equation, $\tilde{h}(x_1, x_2)$.

### 2.4. Implementation Details

All simulations in this paper were completed in COMSOL Multiphysics Version 6.0. The simulations using the three-dimensional model to verify the accuracy of the reduced-dimensional model were performed using the "Laminar Flow" interface. The reduced-dimensional model was implemented using the "Weak Form PDE" interface, with the height field in the reduced-dimensional model created and filtered using the "Optimization" interface. The tolerance for the steady study is $10^{-3}$ and the tolerance for the unsteady study is $5 \times 10^{-3}$.

### 3. Results

#### 3.1. Comparison of Sliced Two-Channel Model

This section compares the results of the full three-dimensional model and the reduced-dimensional model for the sliced two-channel geometry.

This section performs a parametric analysis of the average inlet velocity and the channel heights for the two different models. The velocity is taken from the range $\{0.02, 0.03, 0.04, 0.05\}$ m/s and the channel height in the range of $\{3.5, 4.5, 5.5, 6.5\}$ mm. Figure 8 shows the velocity distribution corresponding to the different parameters. It can be seen from the figure that the reduced-dimensional model is able to simulate the obstruction of flow by the dimples, the effect of channel height on velocity, and the distribution near the outlet of the channel well. However, there is a difference in the velocity distribution around the dimples obtained from the full three-dimensional model and the reduced-dimensional model. This difference is likely caused by the geometric approximation made using the PDE filtered height field in combination with the inherent accuracy imposed by the assumption of a constant out-of-plane velocity profile for the reduced-dimensional model. To quantify the accuracy of the reduced-dimensional model for velocity predictions, this section

compares the velocity magnitudes at the channel outlet between the reduced-dimensional model and the full three-dimensional model and calculates the relative error in velocity, with the specific error values shown in Table 3. Figure 9 shows that the velocity error increases as the channel height increases and decreases as the velocity is increased, but stays below 2.5%. This demonstrates that the reduced-dimensional model has acceptable accuracy for the velocity simulation.

Pressure drop is an important index in fluid simulation, so this section also compares the pressure drop across a single fluid channel obtained using the reduced-dimensional model and the full three-dimensional model. Table 3 shows the pressure drop for both models for different velocities $U_{avg}$ and different channel heights $L_{channel}$. It can be seen that the pressure drop is increasing for larger flow velocities and decreases for larger channel height. This is exactly as expected due to simple fluid resistance principles and was also observed in the experimental tests [1]. It can further be seen that the error between the pressure drops obtained from the reduced-dimensional model and the full three-dimensional model varies within a range of 1% to 10% and that the pressure drops from the reduced-dimensional model are all lower than the pressure drops from the three-dimensional model. This is consistent with the previous conclusion that the height variation of the dimple generated by the PDE filter is more continuous than that of the real dimples and that the edges of the real dimples are sharper, so the pressure drop from the reduced-dimensional model is lower than that of the three-dimensional model. In addition, it can be found that the error in pressure drop shows dependence on $U_{avg}$ and $L_{channel}$. To visualize this, Figure 10 shows the error in the pressure drop of the channel for different inlet velocities and different channel heights. The error in pressure drops decreases significantly with increasing channel height because the effect of the dimple is weakened by increasing channel height when the dimple size is constant. In addition, the error in the pressure drop decreases as the inlet velocity increases. This is because the narrow space of a single dimple provides great resistance to flow, and higher fluid velocities are maintained at intervals between the two dimples, with less fluid dispersed from the main flow into the interior of the single dimple, so the effect of the inaccurate description of the height of the dimple is weakened and the accuracy of the pressure drop obtained from the reduced-dimensional model is improved.

All computations in this section were performed under the same computational conditions (Intel (R) Core (TM) i7-11800H @ 2.30 GHz, 8 cores). The computational costs of the three-dimensional sliced model and the reduced-dimensional sliced model were compared using $U_{in} = 0.2$ m/s and $d_{channel} = 3.5$ mm as examples. The three-dimensional sliced model uses the mesh shown in Figure A1 of Appendix A. When using linear shape functions for both velocity and pressure (P1+P1), 1,779,928 degrees of freedom (DOFs) are solved in 520 s. However, the mesh and interpolation order are very coarse. When quadratic shape functions are used for the velocity (P2+P1), the DOFs to be solved reach 10,302,784. The reduced-dimensional model uses the mesh shown in Figure A3 of Appendix A, which is significantly more refined compared to the three-dimensional mesh. When solved using the P1+P1 strategy, there are 1,103,282 DOFs solved in 38 s, resulting in a velocity profile with more detail than the coarse three-dimensional mesh. When using the P2+P1 strategy, there are 2,973,648 DOFs solved in 177 s. For the full two-channel model in Figure 6, if the full three-dimensional equations were used for the calculation, to obtain an accurate and smooth velocity distribution, the number of DOFs to be solved is estimated to be over a hundred million, which is a great computational challenge.

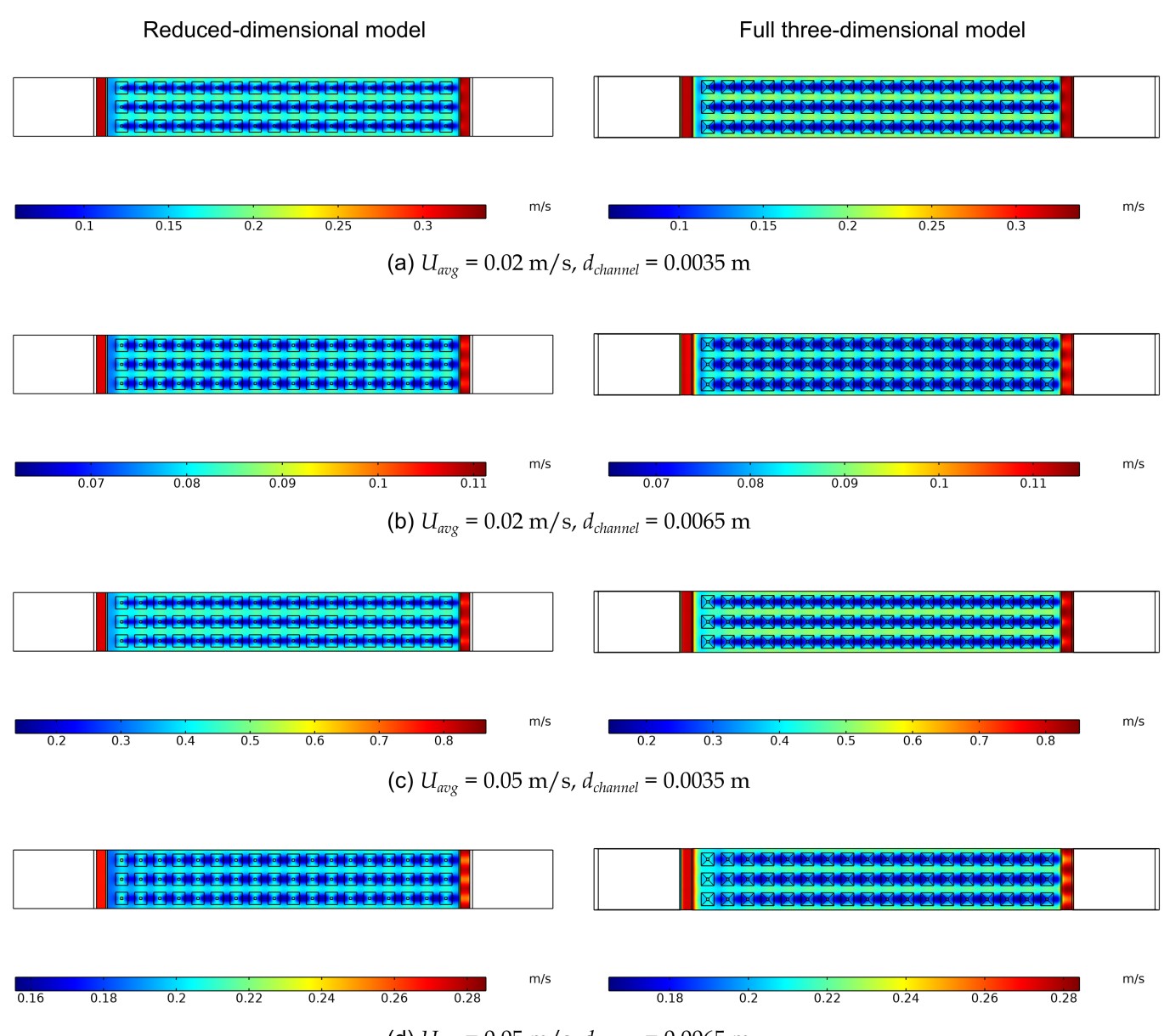

**Figure 8.** Comparison of velocity distributions obtained from full three-dimensional models and reduced-dimensional models for different inlet velocities and different channel thicknesses.

**Table 3.** Comparison of pressure drop for different models with different inlet velocities and different channel heights.

| $U_{avg}$ [m/s] | $d_{channel}$ [mm] | Pressure Drop [Pa] | | | Relative Error of Velocity [%] |
|---|---|---|---|---|---|
| | | 2D | 3D | Error [%] | |
| 0.02 | 3.5 | 0.56 | 0.62 | 9.43 | 0.30 |
| 0.02 | 4.5 | 0.25 | 0.27 | 7.58 | 1.00 |
| 0.02 | 5.5 | 0.13 | 0.14 | 5.90 | 1.81 |
| 0.02 | 6.5 | 0.08 | 0.08 | 4.29 | 2.50 |
| 0.03 | 3.5 | 0.85 | 0.93 | 9.05 | 0.48 |
| 0.03 | 4.5 | 0.37 | 0.40 | 7.13 | 1.09 |
| 0.03 | 5.5 | 0.19 | 0.20 | 5.21 | 1.83 |

**Table 3.** *Cont.*

| $U_{avg}$ [m/s] | $d_{channel}$ [mm] | Pressure Drop [Pa] | | | Relative Error of Velocity [%] |
| --- | --- | --- | --- | --- | --- |
| | | 2D | 3D | Error [%] | |
| 0.03 | 6.5 | 0.11 | 0.12 | 3.26 | 2.47 |
| 0.04 | 3.5 | 1.13 | 1.24 | 8.59 | 0.57 |
| 0.04 | 4.5 | 0.49 | 0.53 | 6.68 | 1.01 |
| 0.04 | 5.5 | 0.26 | 0.27 | 4.55 | 1.56 |
| 0.04 | 6.5 | 0.15 | 0.16 | 2.27 | 2.07 |
| 0.05 | 3.5 | 1.42 | 1.54 | 8.09 | 0.55 |
| 0.05 | 4.5 | 0.62 | 0.66 | 6.25 | 0.74 |
| 0.05 | 5.5 | 0.32 | 0.34 | 3.92 | 1.06 |
| 0.05 | 6.5 | 0.19 | 0.19 | 1.32 | 1.40 |

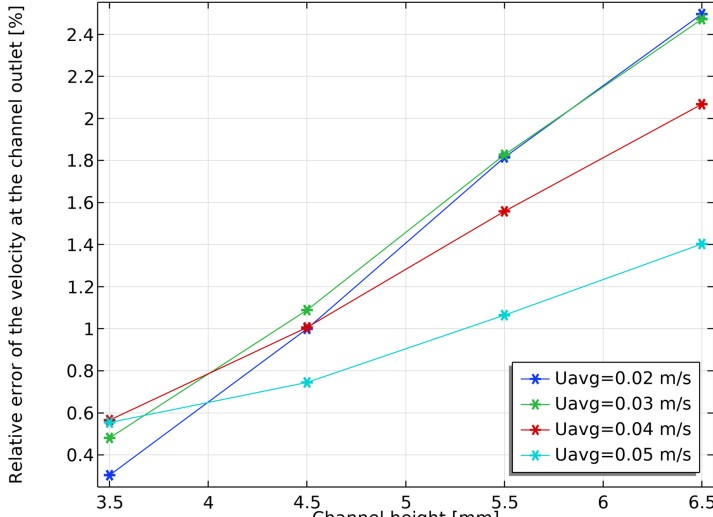

**Figure 9.** Relative error of the velocity at channel outlet between reduced-dimensional model and full three-dimensional model for different inlet velocities and different channel heights.

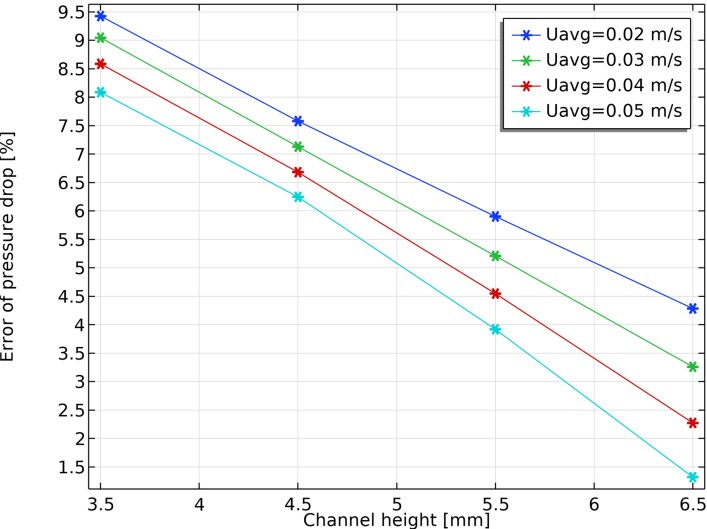

**Figure 10.** Pressure drop error between reduced-dimensional model and full three-dimensional model for different inlet velocities and different channel heights.

*3.2. Full Two-Channel Flow Distribution*

This section shows the velocity distribution of the full two-channel model obtained using the reduced-dimensional model. Furthermore, the effect of the inlet velocity magnitude, the inlet velocity profile, the inlet velocity angle, and the channel height on the velocity distribution are investigated to analyze the influence of the structure on the internal fluid distribution.

### 3.2.1. Reduced-Dimensional Model for Unsteady Flow

Due to the large difference between the channel height and the height of outlet area, the velocity distribution is complex, and it is difficult to reach a steady state. Therefore, the simulation needs to be solved using an unsteady flow model. The reduced-dimensional model for unsteady flow is given by modifying the governing equations with the time-derivative term as follows:

$$\widetilde{\varrho}\frac{\partial \bar{u}_i}{\partial t} + \bar{\varrho}\bar{u}_j\frac{\partial \bar{u}_i}{\partial x_j} - \mu\frac{\partial}{\partial x_j}\left(\frac{\partial \bar{u}_i}{\partial x_j} + \frac{\partial \bar{u}_j}{\partial x_i}\right) + \frac{\partial \bar{p}}{\partial x_i} - \frac{10\mu}{h(\boldsymbol{x})^2}\bar{u}_i = 0 \tag{5a}$$

$$\tilde{h}(\boldsymbol{x})\frac{\partial \bar{u}_i}{\partial x_i} + \bar{u}_i\frac{\partial \tilde{h}(\boldsymbol{x})}{\partial x_i} = 0 \tag{5b}$$

where $\widetilde{\varrho} = \frac{5}{4}\varrho$ is the scaled density for the time derivative. The derivation of the scaled density for the time term, $\widetilde{\varrho}$, is performed in the same manner as the derivation of the scaled density for the convective term, $\bar{\varrho}$. The latter was detailed by Alexandersen [21], so the readers are referred to this for more details.

### 3.2.2. Different Inlet Velocity Profiles

For the physical TES systems in mind [1,2], the symmetry of the velocities flowing into the channels between the plates is unlikely based on unpublished data. In order to study the relative effect of varying asymmetry, a simple nonsymmetric velocity profile is defined as a piecewiese function as follows:

$$u_{in} = \begin{cases} U_{avg}\left(\frac{-y^2+2y_cy}{y_c^2}\right) & \text{for} \quad y < y_c \\ U_{avg}\left(\frac{-y^2+W^2-2y_c(W-y)}{(W-y_c)^2}\right) & \text{for} \quad y \geq y_c \end{cases} \tag{6}$$

where $u_{in}$ is the inlet velocity, $U_{avg}$ is the average magnitude of inlet velocity, $W$ is the width of the inlet, and $y_c$ is the position of the maximum velocity in between $y_c \in [0; W]$. The effect of $U_{avg}$ and $y_c$ on the velocity profile is shown in Figure 11. By altering the parameters $y_c$, the inlet velocity can be easily changed while keeping the inlet flow rate constant.

### 3.2.3. Overall Flow Distribution

The velocity distribution and pressure distribution are shown in Figure 12 for inlet velocity magnitude $U_{avg} = 0.189$ m/s and center-line position of the velocity profile $y_c = W/2$. This specific flow velocity is computed based on a flow rate of 200 m$^3$/h through the full-scale energy storage system shown in Figure 2 and discussed by Dallaire et al. [2].

The velocity increases significantly at the entry to the spacings due to the large decrease in height from the inlet part to the channel. Inside the channel, the velocity between dimples is higher than the velocity inside the dimple. The reason for this is that the height at the dimples is much lower and the resistance to flow is larger, so the internal velocity is lower. In addition, the fluid is impeded by the dimples and forms several parallel flows. The splitting of the flow can be observed more clearly in the outlet area. The two circular areas in the middle of the channel have a lower overall velocity due to their higher height, but due to the higher velocity of the incoming flow formed between the dimples, jets with higher velocities can also be observed in the circular area. Due to the lower resistance to flow, the surrounding fluid is deflected into the circular areas. In addition, the velocity distribution

at the outlet is quite chaotic, and the steady state cannot be reached by extending the simulation time.

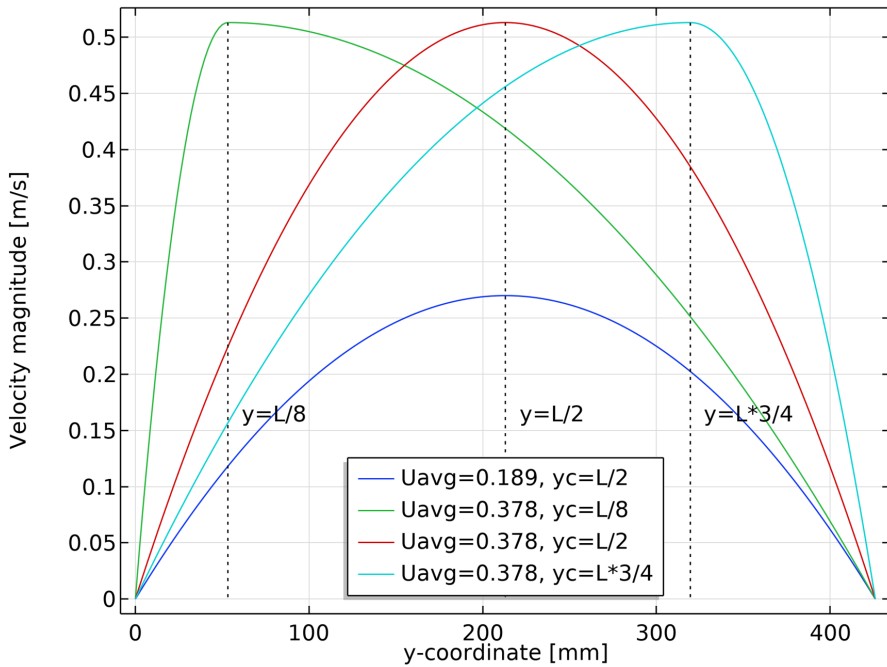

**Figure 11.** Profiles of the inlet velocity for different parameters, where $U_{avg}$ controls the magnitude of the profile and $y_c$ (see Equation (6)) controls the asymmetry of the profile.

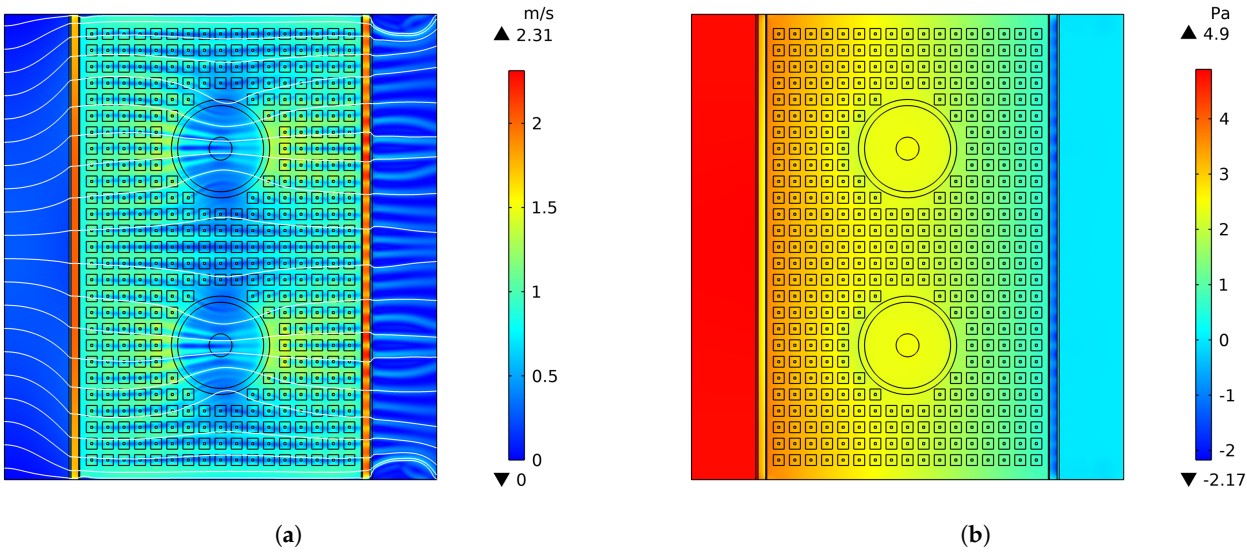

(**a**)      (**b**)

**Figure 12.** Velocity and pressure distributions of the full two-channel model obtained from the reduced-dimensional model when $U_{avg} = 0.189$ m/s and $y_c = W/2$. (**a**) Velocity distribution of full channel; (**b**) pressure distribution of full channel.

At the channel inlet, the pressure drops steeply because the height suddenly contracts. Further on, the height expands slightly and the velocity decreases, so the pressure rises again as it enters the channel. It is worth noting that the pressure distribution shows an intermittent negative distribution at the channel exit, which is due to the series of jets at the outlet, which can be seen in the velocity distribution figure. The jets have locally high velocities, drawing in flow from their surroundings, and therefore the pressure decreases at this point. It can be observed that the location of the high velocity of the jets is consistent with the negative pressure area.

To show the local velocity variation more clearly, different cut lines are made across the fluid channel for plotting the in-plane velocity profiles. Representative locations are shown in Figure 13a. These are the total inlet $\Gamma_{in}^{total}$ and outlet $\Gamma_{out}^{total}$; the inlet $\Gamma_{in}$ and outlet $\Gamma_{out}$ of the channel at very small heights; the location of the dimples $\Gamma_1$, $\Gamma_3$; and the center line of the circular area $\Gamma_2$.

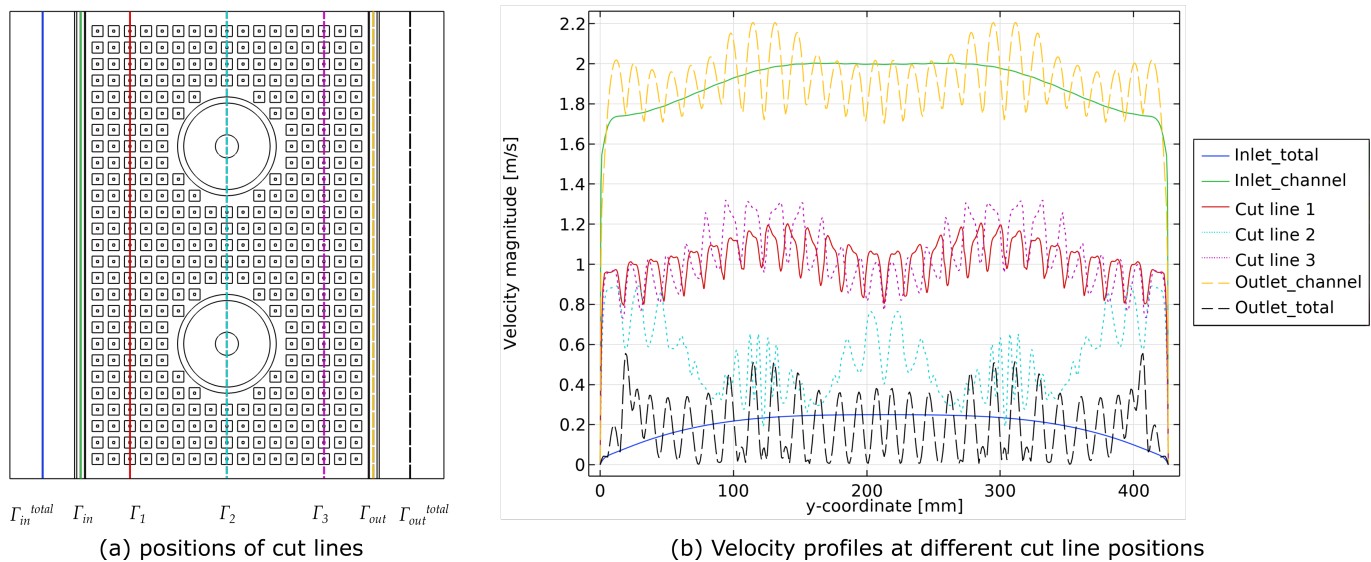

(a) positions of cut lines

(b) Velocity profiles at different cut line positions

**Figure 13.** Velocity profiles at different cut line positions when $y_c = W/2$ and $U_{avg} = 0.189$ m/s.

Figure 13b shows the velocity profiles at the different cut lines when $U_{avg} = 0.189$ m/s. The velocity profiles of $\Gamma_{in}^{total}$ and $\Gamma_{in}$ are smooth, while at the position after the dimples, the velocity profiles show a clear distribution of alternating peaks and valleys. This is due to the high resistance at the dimples, where part of the fluid bypasses the dimples, resulting in low velocity in the center of the dimples and high velocity around the dimples. Although the dimples cause the velocity profile to form peaks and valleys, they also improve the overall uniformity of the velocity. The inlet velocity has a parabolic distribution, with the middle velocity being higher than the velocity near the boundary, while the flow at the outlet is globally more equally distributed, despite the oscillations due to the dimples. This is due to the small channel height and the large number of dimples on the surface, which have a significant resistance to the internal flow, increasing the pressure drop of the fluid and allowing for a more uniform flow distribution, eliminating the nonuniformity of the inlet velocity.

### 3.2.4. Effect of Average Inlet Velocity Magnitude

The effect of the internal structure of the flow channel on the velocity is investigated for different average inlet velocity magnitude $U_{avg} \in \{0.189, 0.378, 0.567\}$ m/s. The magnitude of the inlet velocities are calculated based on the experimental air flow rates (200 m³/h, 400 m³/h, 600 m³/h, respectively), applied to the full-scale energy storage system shown in Figure 2 and discussed by Dallaire et al. [2]. To make the results clearer and the paper more concise, this section only shows the velocity profiles at cut lines of the channel inlet $\Gamma_{in}$ and outlet $\Gamma_{out}$.

Figure 14a shows the velocity profiles at the inlet and outlet of the fluid channel for different velocity magnitudes. The solid lines shows the velocity profile at the inlet and the dashed lines shows the velocity profile at the outlet. For different magnitudes of inlet velocity, the outlet velocity profiles all show a distribution of alternating peaks and valleys. For all velocity profiles, the peaks and valleys are positioned at a fixed location, with the valleys being generated at the center of the dimple. Additionally, the gap between the peaks and valleys in outlet velocity widens with increasing $U_{avg}$, which is intuitive

as the hindering effect of the dimple results in a more pronounced velocity gap as the velocity rises.

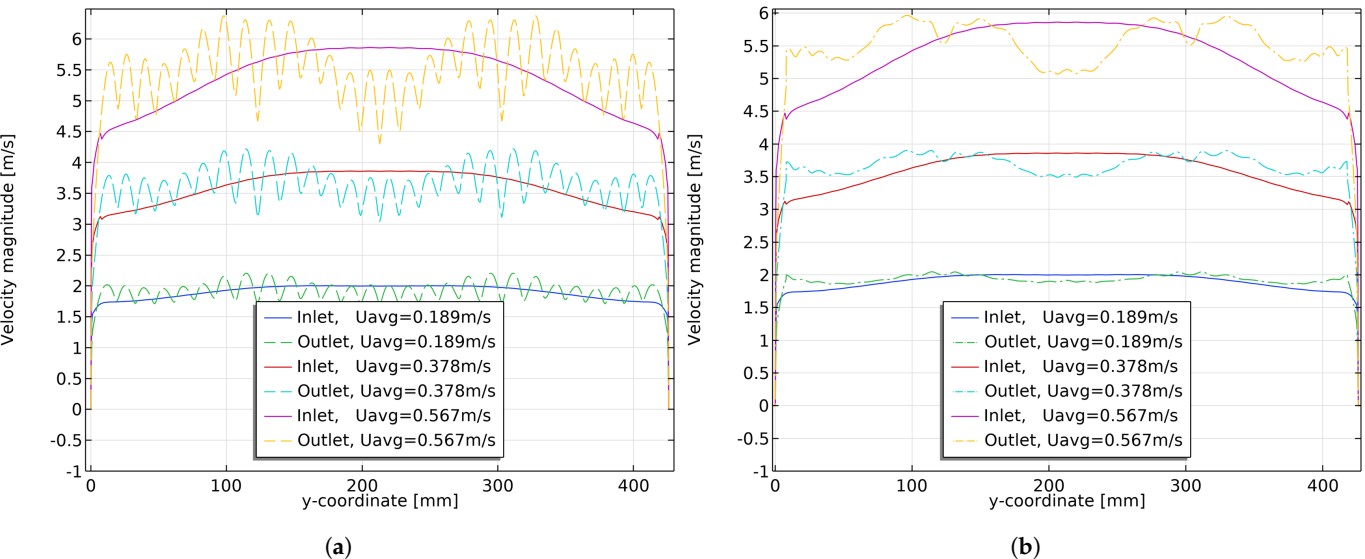

(**a**)                          (**b**)

**Figure 14.** Comparison of velocity profiles over the cut lines at the channel inlet $\Gamma_{in}$ and channel outlet $\Gamma_{out}$ for different velocity magnitudes $U_{avg}$ at $y_c = W/2$. For (**a**), the solid line indicates the inlet velocity and the dashed line indicates the outlet velocity. For (**b**), the solid line indicates the inlet velocity and the dotted line indicates moving average velocity. (**a**) Velocity profiles of the channel inlet and channel outlet; (**b**) channel inlet velocity distribution and the moving average of the channel outlet velocity distributions.

To remove small-scale variations caused by the dimples and allow focus to be placed on the overall flow distribution, a moving average is introduced to filter the peaks and valleys in the velocity profile. The moving average is a common method of data smoothing in statistics, and since there is no ranking relationship between velocities, the simple moving average is used here to smooth the velocity profiles. As the peaks and valleys in the velocity profile are generated due to the obstruction of the dimples, the cycle length of the velocity peaks and valleys corresponds to the distance between the centers of the dimples, so the window length for calculating the moving average is set to the center distance of the dimples. The moving average of the outlet velocity is shown in Figure 14b, where the dash–dotted lines indicate the moving average of the outlet velocity and the solid line indicates the profile of the inlet velocity. After comparing the two, especially for higher average inlet velocities, the overall outlet velocity near the wall is improved, while the velocity at the middle is reduced to be close to the velocity near the wall.

To further analyze the effect of the dimple structure on fluids with different velocity magnitudes, Table 4 shows the mean and standard deviation of the inlet velocity profiles, outlet velocity profiles, and the moving average of the outlet velocity. Firstly, the standard deviation of the velocities shows an overall increasing trend as $U_{avg}$ increases, which means that the outlet velocity becomes less uniform, which is consistent with the previous conclusion. Furthermore, the standard deviation of the moving average of the outlet velocity is significantly lower than that of the inlet velocity, while the standard deviation of the outlet velocity is roughly similar to that of the inlet velocity. This indicates that the overall distribution of outlet velocities is more uniform compared to the inlet velocities but that the uniformity of the local velocity distribution at the outlet becomes worse due to the velocity peaks and valleys generated by the dimple structure.

**Table 4.** Average velocity and standard deviation of the inlet $\Gamma_{in}$ and outlet $\Gamma_{out}$ of the channel and the moving average of the outlet velocity for different velocity magnitudes.

| $U_{avg}$ [m/s] | Average Velocity [m/s] | | | Standard Deviation [-] | | |
|---|---|---|---|---|---|---|
| | Inlet | Outlet | Mov. avg. | Inlet | Outlet | Mov. avg. |
| 0.189 | 1.9345 | 1.9367 | 1.9364 | 0.0998 | 0.1259 | 0.0528 |
| 0.378 | 3.6759 | 3.6951 | 3.6953 | 0.2590 | 0.2658 | 0.1295 |
| 0.567 | 5.5148 | 5.5600 | 5.5612 | 0.4778 | 0.4466 | 0.2709 |

### 3.2.5. Effect of Asymmetry in Inlet Velocity

The effect of the dimple structure within the fluid channel on fluids with different inlet asymmetry parameters $y_c$ is also investigated in this section. The average magnitude of the velocity $U_{avg} = 0.189$ m/s, and the parameter $y_c \in \{\frac{W}{8}, \frac{W}{4}, \frac{W}{2}, \frac{3W}{4}, \frac{7W}{8}\}$, while the other parameters remain unchanged. In particular, it is noted that the flow rate of the inlet remains constant. The velocity profile of the inlet is shown in Figure 15a. The difference between the inlet velocity profiles for different $y_c$ is significant: when $y_c = \frac{W}{2}$, the velocity distribution is symmetrical; when $y_c$ departs from $\frac{W}{2}$, the velocity distribution is asymmetrical and uneven. However, the outlet velocity exhibits an extreme insensitivity to variations in the inlet velocity distribution, with the outlet velocity distribution obtained for different inlet velocities overlapping. This suggests that the thin spacing together with the dimple structure significantly changes the flow of the fluid, eliminating the difference in inlet flow distribution.

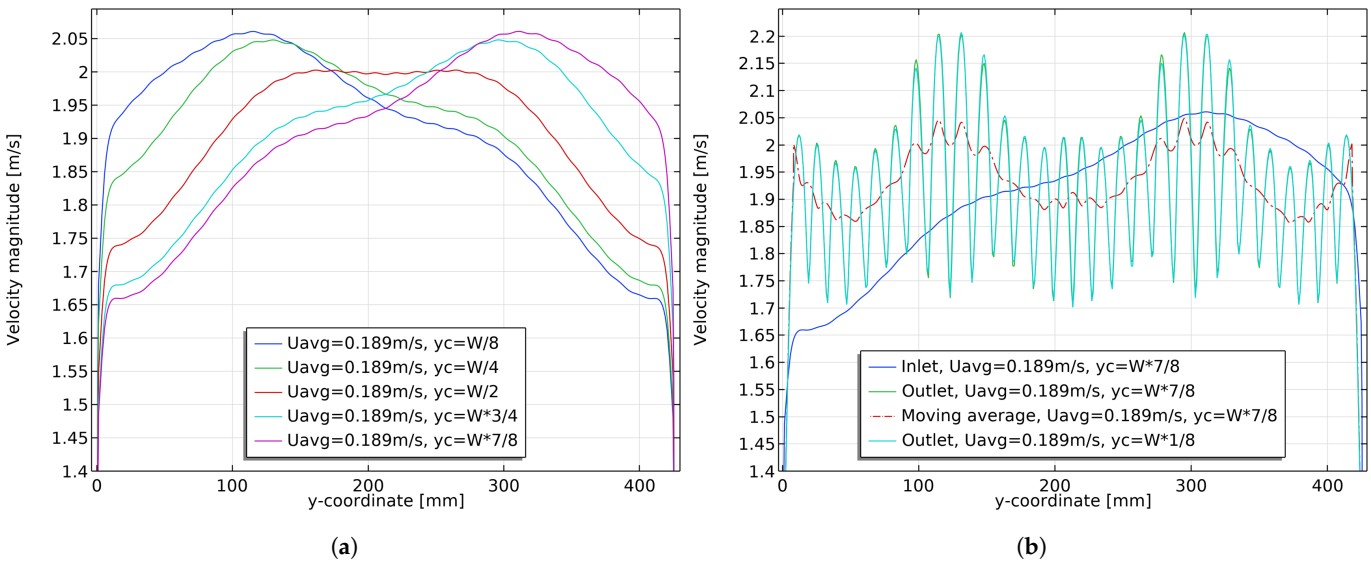

(**a**)   (**b**)

**Figure 15.** Velocity profiles of the channel inlet $\Gamma_{in}$ and channel outlet $\Gamma_{out}$ for different $y_c$ when $U_{avg} = 0.189$ m/s. (**a**) Velocity profiles at the channel inlet for different $y_c$; (**b**) channel inlet and outlet velocity profiles when $y_c = \frac{7W}{8}$.

Taking the velocity distribution at $y_c = \frac{7W}{8}$ as representative, Figure 15b shows the inlet velocity, the outlet velocity, and the corresponding moving average. The figure shows that the outlet velocity profile is still distributed with alternating peaks and valleys, which makes the local uniformity of the outlet velocity worse. The moving average of the outlet velocity, on the other hand, shows that the overall distribution is more uniform compared to the distribution of the inlet velocity.

Table 5 shows the mean and standard deviation of inlet and outlet velocities and moving averages of outlet velocities. The standard deviation of the moving average of the outlet velocity is significantly smaller than that of the inlet velocity for all groups, which confirms that the overall distribution of the outlet velocity is better than the inlet velocity.

The standard deviation of the outlet velocity is close to or even higher than the standard deviation of the inlet velocity, which is due to the local uneven distribution of the outlet velocity affecting the overall standard deviation. In addition, the table shows that the standard deviation of the inlet velocity varies significantly with $y_c$, while the standard deviation of the outlet velocity remains almost constant, which confirms that the outlet velocity shows a strong insensitivity to variations in the inlet velocity.

**Table 5.** Average velocity and standard deviation of the inlet $\Gamma_{in}$ and outlet $\Gamma_{out}$ of the channel and the moving average of the outlet velocity for different $y_c$ when $U_{avg} = 0.189$ m/s.

| $y_c$ [mm] | Average Velocity [m/s] | | | Standard Deviation [-] | | |
|---|---|---|---|---|---|---|
| | Inlet | Outlet | Mov. avg. | Inlet | Outlet | Mov. avg. |
| W/8 | 1.9392 | 1.9368 | 1.9365 | 0.1254 | 0.1252 | 0.0529 |
| W/4 | 1.9366 | 1.9367 | 1.9364 | 0.1115 | 0.1256 | 0.0529 |
| W/2 | 1.9345 | 1.9367 | 1.9364 | 0.0998 | 0.1259 | 0.0528 |
| 3W/4 | 1.9366 | 1.9366 | 1.9364 | 0.1113 | 0.1257 | 0.0530 |
| 7W/8 | 1.9392 | 1.9367 | 1.9364 | 0.1251 | 0.1253 | 0.0530 |

3.2.6. Effect of Inlet Velocity Angles

In the novel air flow configuration dual-stack module tested by Dallaire et al. [2], the fluid flows into the gaps at an angle due to the space constraints of the equipment, so this section also investigates the effect of different inlet angles on the outlet velocity distribution. The expressions for the *x*-directional component of the inlet velocity *u* and *y*-directional component of the inlet velocity *v* are as follows:

$$u = u_{in}$$
$$v = u_{in} \tan \theta \tag{7}$$

where $\theta$ is the inlet angle of the fluid and $\theta \in \{0, \frac{\pi}{6}, \frac{\pi}{5}, \frac{\pi}{4}, \frac{\pi}{3}\}$. The components of the inlet velocity in each direction are shown in Figure 16.

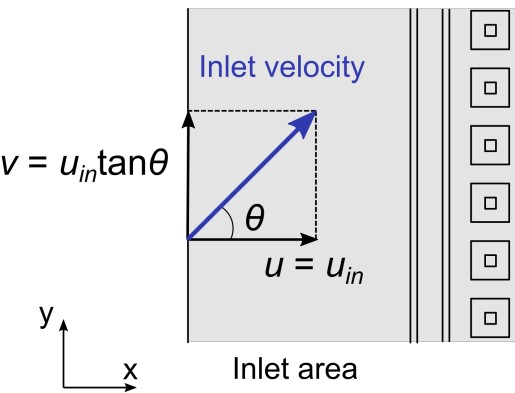

**Figure 16.** Schematic diagram of the angle of inlet velocity and the velocity component.

Letting $y_c = \frac{W}{2}$ and $U_{avg} = 0.189$ m/s to ensure a constant inlet flow rate, the velocity profile at inlet $\Gamma_{in}$ is shown in the Figure 17a. The inlet angle has a strong effect on the inlet velocity profile: when $\theta = 0$, the velocity distribution is symmetrical; when the angle increases, the velocity distribution is no longer symmetrical and the velocity at one side increases. However, the outlet velocity is not sensitive to variations in the inlet angle, and all the outlet velocity profiles almost overlap. Taking $\theta = \frac{\pi}{3}$ as an example, Figure 17b shows the inlet velocity, the outlet velocity, and the moving average of the outlet velocity. From the figure, it can be concluded that the thin spacing and dimple structure eliminates the asymmetric effect of the inlet velocity, yielding alternating peak and valley velocity

distributions, whose overall uniformity of velocity becomes better but local uniformity becomes worse.

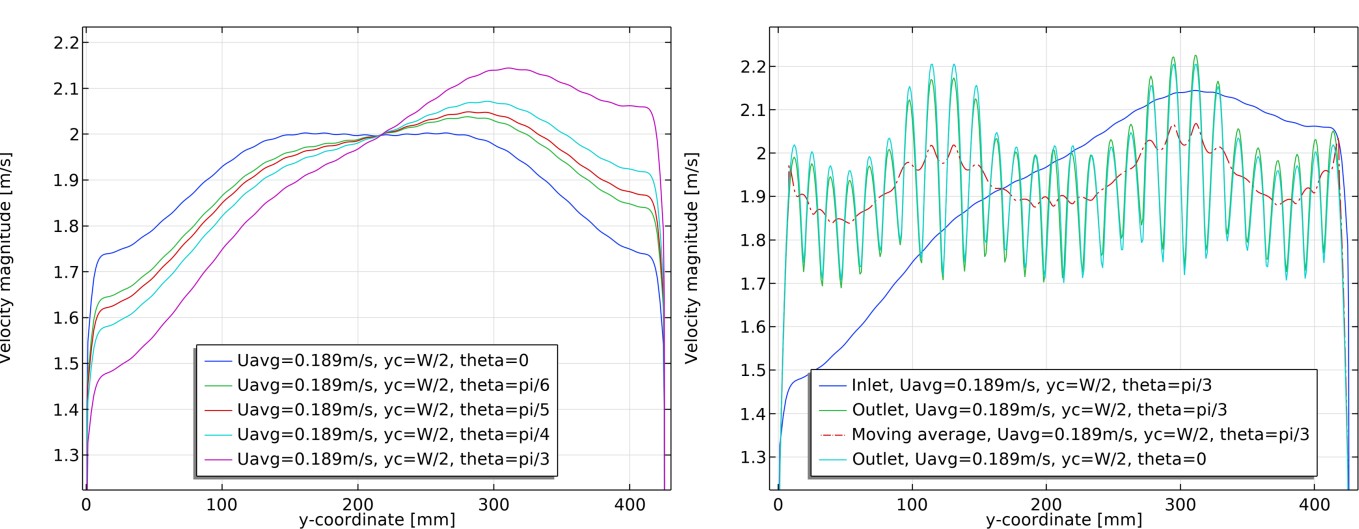

(**a**)                (**b**)

**Figure 17.** Velocity profiles of channel inlet $\Gamma_{in}$ and channel outlet $\Gamma_{out}$ for different velocity angles $\theta$ when $U_{avg} = 0.189$ m/s and $y_c = W/2$. (**a**) Velocity profiles at channel inlet for different velocity angles $\theta$; (**b**) channel inlet and outlet velocity profiles when $\theta = \frac{\pi}{3}$.

Table 6 shows the mean and standard deviation of inlet and outlet velocities and moving averages of outlet velocities. The standard deviation of the inlet velocity increases as $\theta$ increases, which is obvious as the uniformity of the inlet velocity becomes worse as the angle increases. However, the standard deviation of the outlet velocity and the moving average of the outlet velocity remain almost constant, which means that the outlet velocity is insensitive also to variations in the angle of the inlet velocity. The standard deviation of the moving average of the outlet velocity is significantly smaller than the standard deviation of the inlet velocity, which means that the overall uniformity of the outlet velocity is improved compared to the inlet velocity. Especially when the inlet angle is large and the velocity uniformity is poor, the thin spacing and dimple structure can significantly improve the uniformity of the outlet velocity.

**Table 6.** Average velocity and standard deviation of the inlet $\Gamma_{in}$ and outlet $\Gamma_{out}$ of the channel and the moving average of the outlet velocity for different $\theta$ when $U_{avg} = 0.189$ m/s, $y_c = W/2$.

| $\theta$ [rad] | Average Velocity [m/s] | | | Standard Deviation [-] | | |
|---|---|---|---|---|---|---|
| | Inlet | Outlet | Mov. avg. | Inlet | Outlet | Mov. avg. |
| 0 | 1.9345 | 1.9367 | 1.9364 | 0.0998 | 0.1259 | 0.0528 |
| $\pi/6$ | 1.9358 | 1.9367 | 1.9364 | 0.1181 | 0.1258 | 0.0534 |
| $\pi/5$ | 1.9366 | 1.9367 | 1.9364 | 0.1277 | 0.1256 | 0.0536 |
| $\pi/4$ | 1.9385 | 1.9366 | 1.9363 | 0.1485 | 0.1255 | 0.0542 |
| $\pi/3$ | 1.9467 | 1.9366 | 1.9363 | 0.2149 | 0.1256 | 0.0562 |

### 3.2.7. Effect of Plate Spacing

The previous section investigated the velocity distribution in fluid channels between parallel plates with a 3.5 mm spacing for different inlet velocities. The results show that the structure of the dimples inside the channel changes the fluid flow significantly. The dimples with very small spacing have large resistance, which eliminates the difference in inlet velocity and results in uniform outlet velocity profiles with alternating peaks and valleys. Inspired by the studies on the standalone system by Hassan et al. [1], the effect of plate spacing on the velocity distribution is investigated.

In this section, the nominal spacing between the parallel plates is increased to 6.5 mm and the dimensions of the dimples are kept constant since the plates are the same. To ensure the same flow rate at the inlet, the average inlet velocity becomes $U_{avg} = 0.159$ m/s. The new system is analyzed for inlet velocity profile parameter $y_c \in \{\frac{W}{8}, \frac{W}{4}, \frac{W}{2}, \frac{3W}{4}, \frac{7W}{8}\}$ and inlet angle $\theta \in \{0, \frac{\pi}{6}, \frac{\pi}{5}, \frac{\pi}{4}, \frac{\pi}{3}\}$.

Figure 18 shows the velocity distribution and pressure distribution when $y_c = W/2$ and $\theta = 0$, which is quite different from the previous velocity distribution. Firstly, the overall velocity magnitude is lower than the previous velocity magnitude due to the increasing height of the channel. Secondly, the uniformity of the velocity distribution becomes worse: at the inlet, the velocity in the middle of the channel is significantly higher than that near the wall; inside the channel, the velocity in the $y$-direction at locations in front of and behind the circular area is significantly higher than at other locations; the velocity at locations corresponding to the circular area in outlet areas $\Gamma_{out}$ and $\Gamma_{out}^{total}$ is also higher than at other locations. In addition, there is no obvious bypassing around the dimples, and the velocity difference between the center of the dimples and the area outside the dimples is not significant. This makes sense as the channel spacing increases, the viscous forces on the fluid decrease, the inertial effects on the flow increase, and the fluid prefers to maintain the initial velocity distribution. Additionally, the resistance between dimples decreases, and therefore the loss of velocity in the dimples is also reduced, which explains why there is no significant velocity difference between the center of the dimples and the surrounding area. The pressure drop in Figure 18b is also seen to be significantly lower than for a spacing of 3.5 mm, Figure 12b, which is as expected due to less flow resistance, and confirms the observations by Hassan et al. [1].

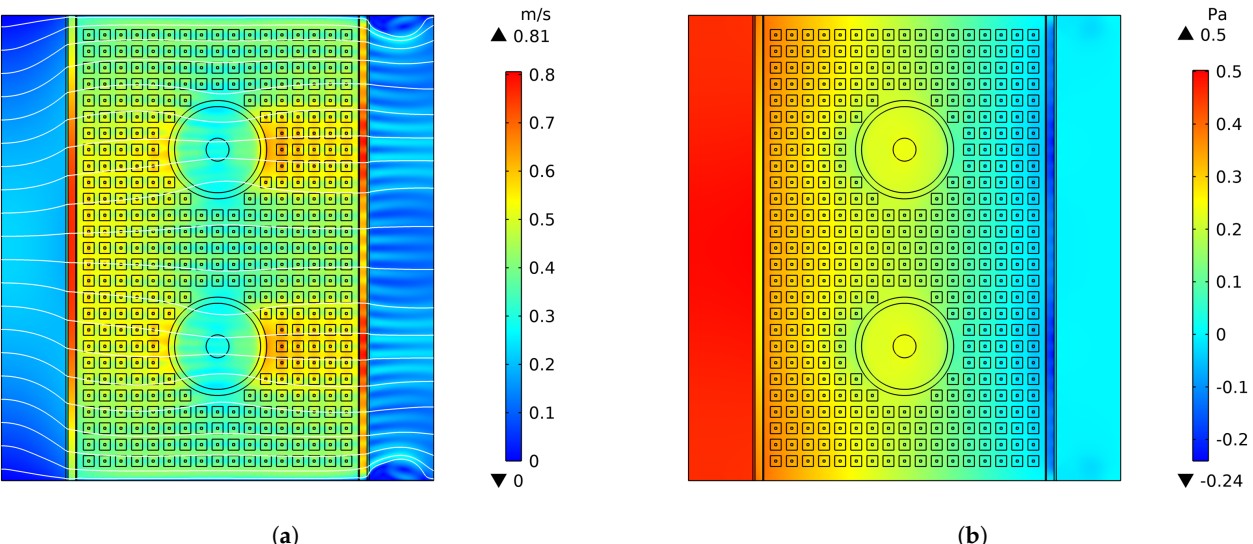

(**a**) (**b**)

**Figure 18.** Velocity and pressure distributions of the full channel when $U_{avg} = 0.159$ m/s, $y_c = W/2$, $\theta = 0$, and the channel spacing height is 6.5 mm. (**a**) Velocity distribution of full channel; (**b**) pressure distribution of full channel.

The velocity profiles at the cut lines are plotted in Figure 19. As can be seen in the figure, the overall uniformity of the large-spacing channel is worse than that of the small-spacing channel. In addition, the velocity profile within the fluid channel still maintains the distribution of alternating peaks and valleys, but the velocity gap between adjacent peaks and valleys becomes smaller because the dimples present less of an obstacle relative to the large spacing.

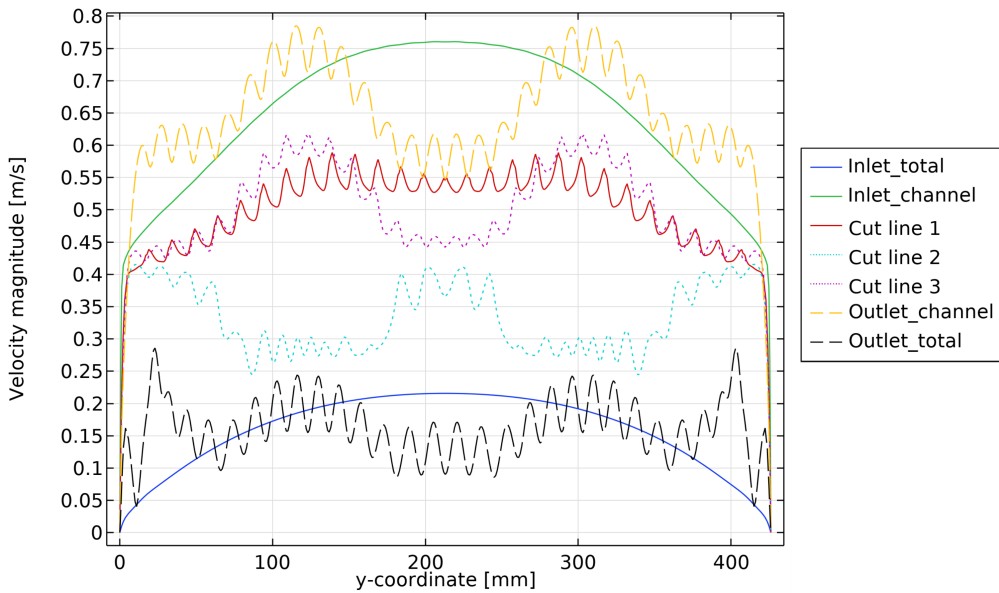

**Figure 19.** Velocity profiles at different cut line positions when $\theta = 0$, $y_c = \frac{W}{2}$, $U_{avg} = 0.159$ m/s, and the channel spacing height is 6.5 mm.

To investigate the effect of increasing channel spacing on the velocity distribution, different velocity profiles and different inlet angles are considered for $U_{avg} = 0.159$ m/s. Let $y_c \in \{\frac{W}{8}, \frac{W}{4}, \frac{W}{2}, \frac{3W}{4}, \frac{7W}{8}\}$ and $\theta \in \{0, \frac{\pi}{6}, \frac{\pi}{5}, \frac{\pi}{4}, \frac{\pi}{3}\}$.

Figure 20 shows the velocity profiles at $\Gamma_{out}$ for different values of $y_c$ and $\theta$. For both cases, it can be seen that the internal structure of the fluid channel can still eliminate the effect of the inlet velocity, forming approximately similar velocity distributions at the outlet. However, increasing the spacing makes the effect of the dimples on the internal velocity distribution weaker, and the influence of the inlet velocity is not completely eliminated. The outlet velocity distribution is influenced by the inlet velocity distribution, with one side of the center line having a higher velocity than the other, and the asymmetry of the outlet velocity increases as the asymmetry of the inlet velocity increases.

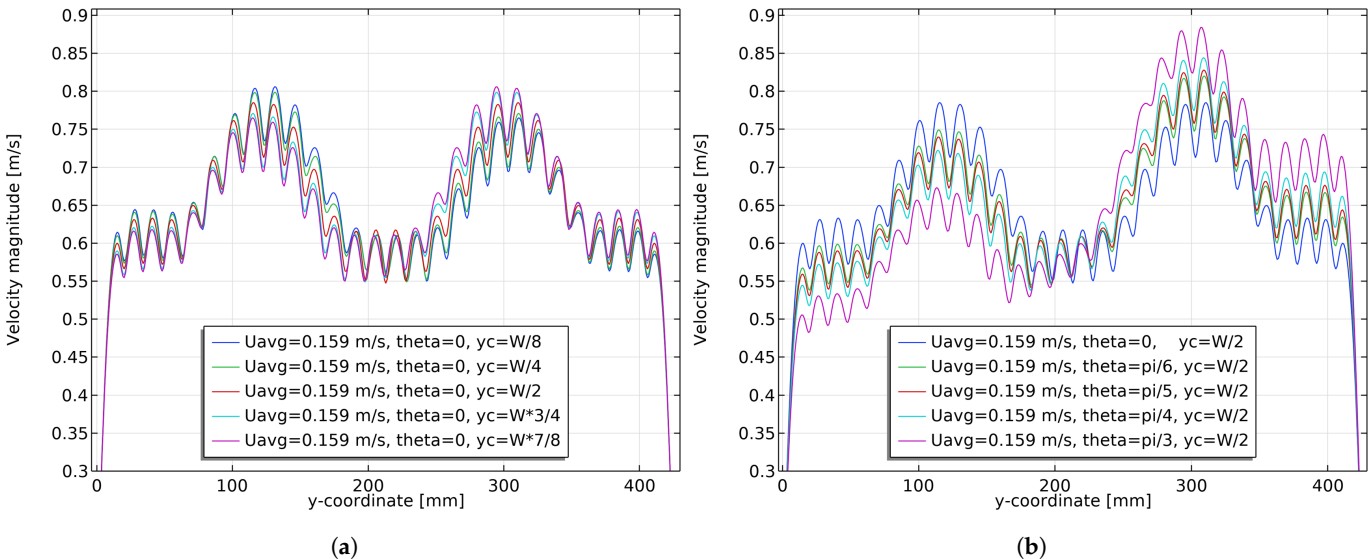

(**a**)　　　　　　　　　　　　　　　　(**b**)

**Figure 20.** Velocity profiles of the channel outlet $\Gamma_{out}$ for different $y_c$ when $\theta = 0$ and different $\theta$ when $y_c = \frac{W}{2}$. The channel spacing is 6.5 mm and the average inlet velocity is $U_{avg} = 0.159$ m/s. (**a**) Different $y_c$; (**b**) different $\theta$.

Hassan et al. [1] concluded that the smaller spacing, and thus channel height, between plates was the preferable option from an energy storage point of view. This is in part based on the fact that more plates can be stacked into the same volume with smaller spacings. The present work indicates that the smaller spacing also helps even out the overall flow distribution over the plates, likely yielding a better utilization of the entire storage potential per plate.

## 4. Conclusions

This work demonstrates the accuracy of an extended reduced-dimensional model with a multichannel height distribution strategy through the analysis of a sliced two-channel model. Furthermore, the work applies the above method to the analysis of the full two-channel model to investigate the effect of the plate topography on the internal fluid velocity distribution for varying external parameters such as asymmetry.

By comparing the velocity and pressure drop of a sliced two-channel model obtained from the reduced-dimensional model with those obtained from the full three-dimensional model, it can be concluded that the reduced-dimensional model combined with the multichannel height distribution strategy can achieve satisfactory accuracy in terms of velocity and pressure drop. However, the accuracy of the method is affected by the height variation and the magnitude of velocity. The velocity error at the channel outlet is limited to 2.5%, and the pressure drop error is limited to 10%. For the pressure drop error, the height of the channel has a greater effect on the error than the magnitude of the inlet velocity. As the topography of the plate surface remains constant, increasing the channel height means that the effect of the topography is reduced, and the pressure drop is more accurately modeled. For velocity errors, both velocity magnitude and channel height have a small effect, and both seem to increase or decrease the error relatively by changing the overall velocity level. The above conclusions are valid for the topography of the plate considered in this work. The application of the method still requires analysis of different parameters to ensure that the model accurately reflects the current problem and experimental validation to ensure the model accurately reflects the physical problem.

The verified reduced-dimensional model is used to perform a parametric analysis of the full two-channel model to investigate the effect of the plate topography on the internal fluid velocity distribution under varying outer conditions governing asymmetry. For flows with different velocity magnitudes and at different incidence angles, the dimples changed the flow, resulting in the same alternating peak-to-valley distribution of the outlet velocity. The smaller the channel height, the less sensitive the outlet velocity profile is to the variation of the angle and asymmetry of the inlet velocity profile. In addition, the plate topography and the small channel height help to improve the overall uniformity of the outlet velocity independent of inflow asymmetry and angle, but an alternating peak–valley distribution of the outlet velocity will have a worse local uniformity.

This work applies the reduced-dimensional model with multichannel height distribution to a practical engineering problem and shows satisfactory accuracy. The observations made from the studies herein provide confidence in the TES module designed in the project [1–3]. However, there are still many aspects that can be improved for practical engineering problems and even the considered application itself. Herein, only relatively low mass flows are considered, whereas the real system is tested for significantly larger ones. That means that the present work only considers flows with small to moderate Reynolds numbers, and future work should consider turbulence effects to better simulate the real flow conditions. In addition, the present work only investigates the fluid flow characteristics in the plate stacks; future work should further investigate the internal heat transfer characteristics.

**Author Contributions:** Conceptualization, J.A.; methodology, J.A.; software, H.M.A.H. and Y.S.; formal analysis, Y.S.; investigation, H.M.A.H. and Y.S.; resources, J.A.; writing—original draft preparation, H.M.A.H. and Y.S.; writing—review and editing, J.A. and Y.S.; visualization, Y.S.; supervision, J.A.; project administration, J.A. All authors have read and agreed to the published version of the manuscript.

**Funding:** The work by J.A. and H.M.A.H. was partially sponsored by the "NeGeV: Next Generation Ventilation" project funded by the Danish Energy Agency under the Energy Technology Development and Demonstration Program (EUDP project number 64017-05117). The authors acknowledge Ivar Lund and Christian Veje for providing the funding for the study. The work by Y.S. was performed during extended stay at University of Southern Denmark funded by the Study Abroad Program from Central South University and China Scholarship Council.

**Data Availability Statement:** An example COMSOL file for the reduced-dimensional model for the sliced two-channel geometry is available on GitHub: https://github.com/sdu-multiphysics/topography/blob/main/application/Sliced_two_channel.mph. Further information for reproduction is available by contacting the corresponding author.

**Conflicts of Interest:** The authors declare no conflict of interest.

## Abbreviations

The following abbreviations are used in this manuscript:

| | |
|---|---|
| CSM | Compact Storage Module |
| DOFs | Degrees of freedom |
| PCM | Phase change material |
| TES | Thermal energy storage |

## Appendix A. Mesh Convergence Analysis

This appendix provides proof of mesh convergence for all simulations in this study. This section shows and verifies the convergence of the three-dimensional and two-dimensional meshes used for the simulation of the sliced two-channel model, and the two-dimensional meshes used for the simulation of the full two-channel model.

### Appendix A.1. Sliced Two-Channel Model

This sections performs a parametric analysis of the sliced two-channel model using a full three-dimensional model and a reduced-dimensional model, respectively. The mesh of the three-dimensional model varies with the geometric parameters; here, the parameter set of $U_{avg} = 0.05$ and $d_{dimple} = 1.5$ is chosen as an example to demonstrate the validity of the mesh.

The mesh used for the three-dimensional sliced model is shown in Figure A1. It consists mainly of tetrahedral elements, with a more accurate mesh for the inlet and outlet surfaces and the channel; the interior of the channel is generated by sweeping to ensure that there is sufficient mesh even at very small distances; the total number of elements in this mesh is 878,816.

The pressure drop between the total inlet and total outlet is taken as a measure of the validity of the mesh, while the number of elements is expressed in terms of the DOFs. As can be seen from the blue line in Figure A2, the pressure drop of the model gradually decreases as the number of degrees of freedom increases, eventually converging. After considering the computational cost and accuracy, the simulation of the three-dimensional sliced two-channel model was performed using a mesh with 878,816 elements yielding 1,779,928 DOFs.

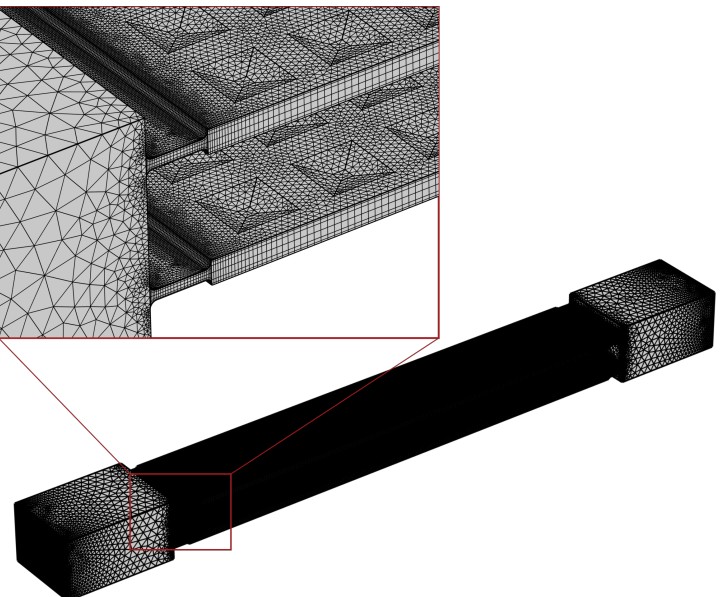

**Figure A1.** Illustration of the three-dimensional mesh used for the sliced two-channel model.

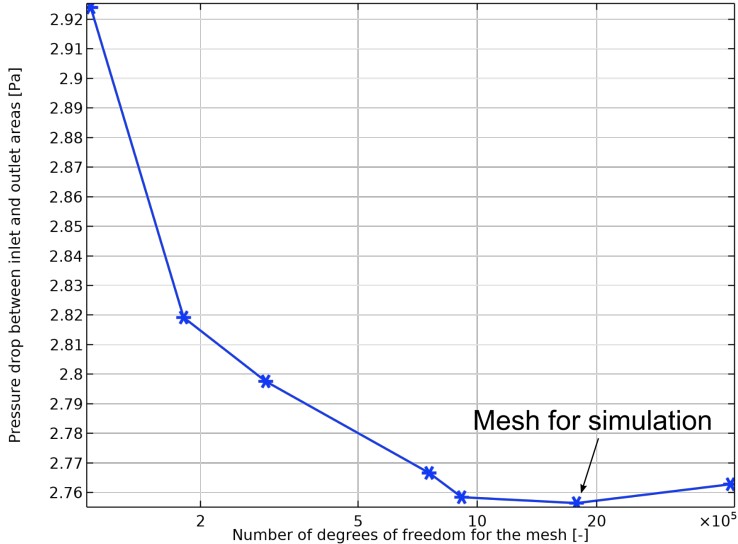

**Figure A2.** The pressure drop of the three-dimensional sliced two-channel model varies with the number of DOFs.

The mesh used for the reduced-dimensional model is partially shown in Figure A3. It consists entirely of triangular elements and has a total of 573,238 elements. Finer meshes are used at the inlet and outlet of the channel and at the dimple, which can be seen to be finer than the mesh for the three-dimensional model. Based on the layout of this mesh, the mesh convergence is investigated by changing the size of the elements to adjust the number of elements.

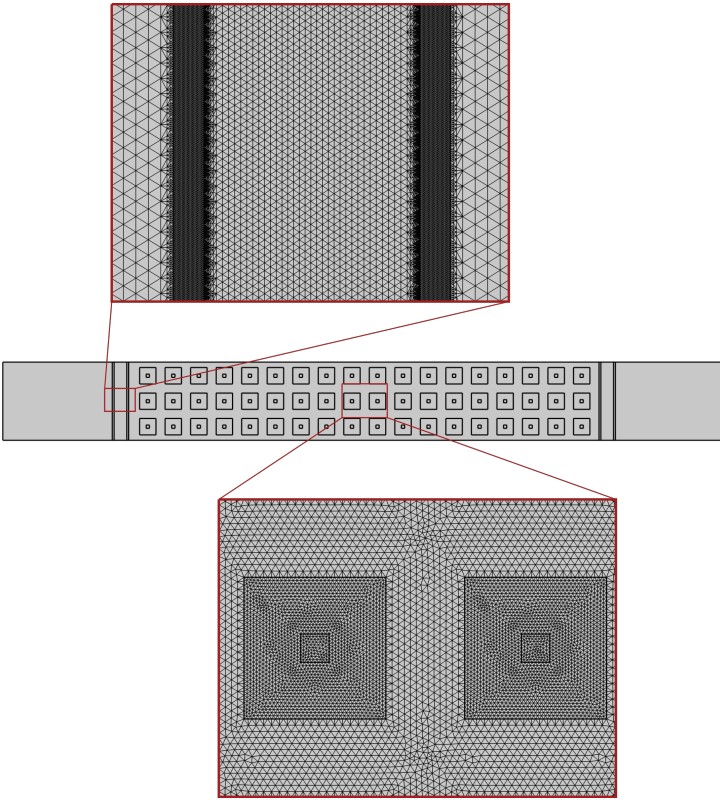

**Figure A3.** The mesh for the sliced two-channel model for reduced-dimensional model.

The pressure drop between the inlet and outlet is taken as a measure of the effectiveness of the mesh, and the measures varied with DOFs as shown in Figure A4. The pressure drop of the model decreases as the number of degrees of freedom increases and eventually converges. After considering the computational cost and accuracy, the simulation of the reduced-dimensional was performed using a mesh with 573,238 elements, yielding 2,973,648 DOFs (including 166,052 additional DOFs introduced by the height field and filter) to be solved using a P2+P1 discretization strategy.

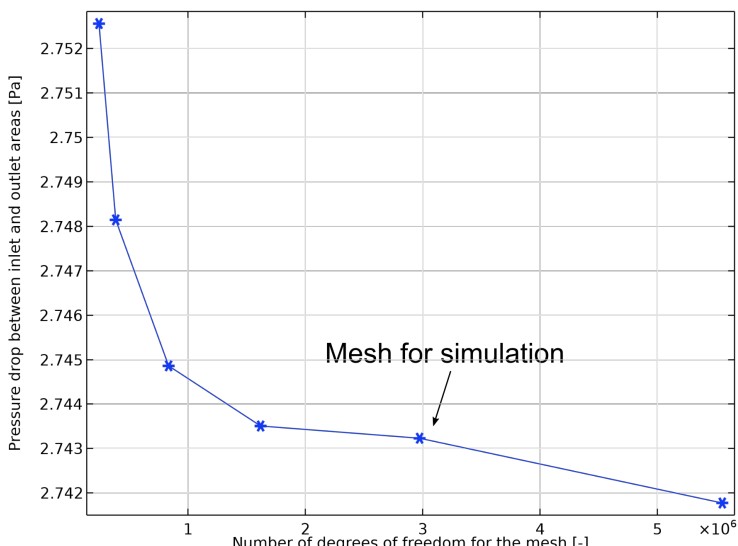

**Figure A4.** The pressure drop of the reduced-dimensional model varies with the number of DOFs.

### Appendix A.2. Full Two-Channel Model

This section presents a comprehensive parametric analysis of a full two-channel model using a reduced-dimensional model. The mesh of the model varies with the geometric parameters, and the parameter sets of $U_{avg} = 0.189$ m/s and $d_{dimple} = 1.5$ mm are chosen here as examples to demonstrate the validity of the mesh.

The mesh used for the reduced-dimensional model is shown in Figure A5. It consists mainly of triangular elements, and a finer mesh is used for the inlet and outlet of the channel to ensure accurate calculation of the effect of height variation on fluid flow. The total number of elements in this mesh is 400,242. The pressure drop between the total inlet and the total outlet is used as a measure of the effectiveness of the mesh, while the number of elements is expressed in terms of the DOFs to be solved.

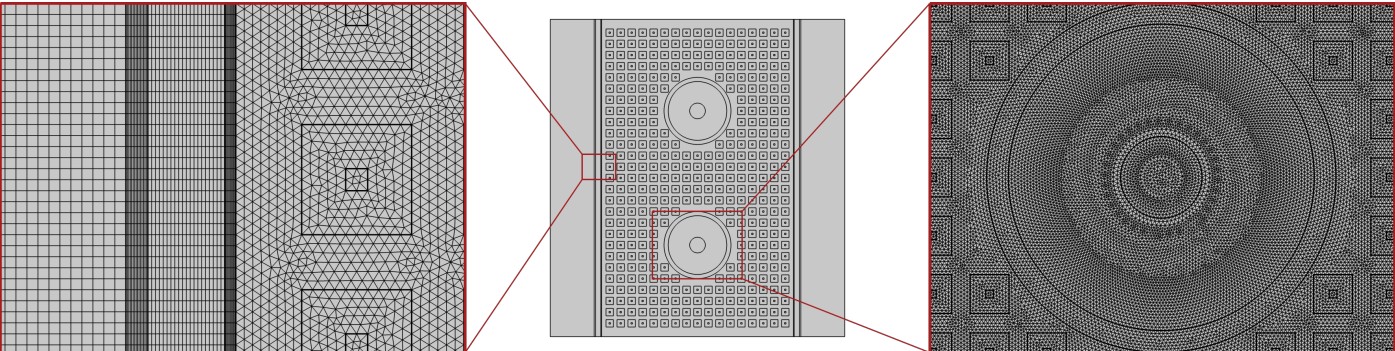

**Figure A5.** The mesh for full two-channel model for reduced-dimensional model.

Figure A6 illustrates the variation of pressure drops between the inlet and outlet with DOFs, from which it can be seen that as the number of meshes increases, the pressure drop of the model gradually decreases and gradually achieves convergence. In consideration of the computational cost and the accuracy of the model, the mesh with 400,242 elements yielded 2,325,861 DOFs (including 102,627 additional DOFs introduced by the height field and filter).

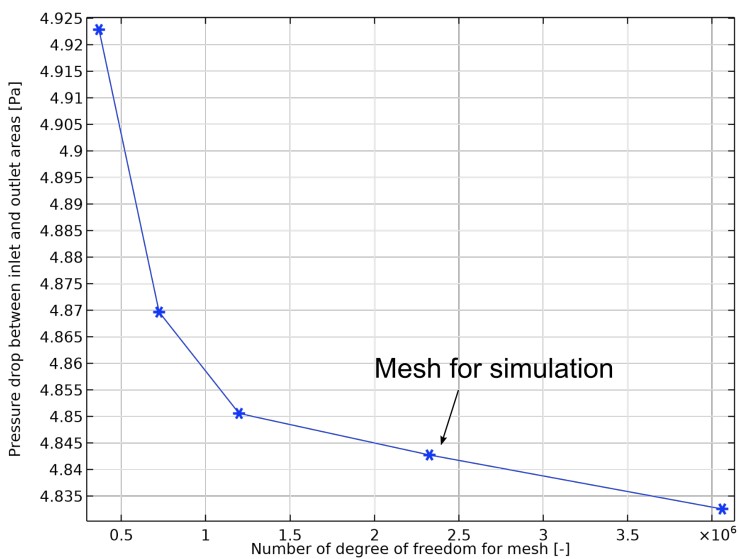

**Figure A6.** The pressure drop of the reduced-dimensional model varies with the number of DOFs.

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
