# Peer review of "Application of a Reduced-Dimensional Model for Fluid Flow between Stacks of Parallel Plates with Complex Surface Topography"

_fluids, doi:10.3390/fluids8060174_

Round 1

Reviewer 1 Report

I suggest the authors seriously consider the following encouraging points to further improve the paper presentation and content;

1.      The Abstract should contain answers to the following questions: What problem was studied, and why is it important? What methods were used? What are the important results? What conclusions can be drawn from the results? What is the novelty of the work, and where does it go beyond previous efforts in the literature? Please include specific and quantitative results in the Abstract while ensuring that it is suitable for a broad audience. References, figures, tables, equations, and abbreviations should be avoided like “(J. Alexandersen, Topography optimisation of fluid flow between parallel plates of spatially-varying spacing: Revisiting the origin of fluid flow topology optimisation, Structural and Multidisciplinary Optimization 65 (2022) 152. doi:10.1007/s00158-022-03243-8)

2.      The authors should try to give advantages of using their method compared to others.

3.      The governing equations and boundary conditions should be given suitable references.

4.      Grammatical and punctuation errors are should be revised carefully. Improve the result and discussion section physically.

5.       All the symbols should be defined clearly

6.      The conclusion section needs revision. Comment: Start the conclusion section with a sentence stating the aim of the study elaborately. Then remark that the major objectives had been established. Start the conclusion section with a fact on the achievement of the research aim before stating conclusive statements. Revise the title to present the aim concisely. The aim of the title should reflect in the abstract, and at the beginning of the conclusion. Author's should revise the conclusion section to provide conclusive statements on the research questions posed at the end of the introduction. Try to itemize all the conclusive facts.

7.      Discussion should be based on the physical reasoning of the model, explaining only the behavior of the curves is not enough. One can easily study the behavior through curves, the important is how this will affect the overall model. The results must be supported by some quantitative analysis.

8.      An updated and complete literature review should be conducted and should appear as part of the Introduction, Radiative Heat And Mass Transfer Analysis Of Micropolar Nanofluid Flow Of Casson Fluid Between Two Rotating Parallel Plates With Effects Of Hall Current, Three dimensional third grade nanofluid flow in a rotating system between parallel plates with Brownian motion and thermophoresis effects, The electrical MHD and hall current impact on micropolar nanofluid flow between rotating parallel plates

Grammatical and punctuation errors are should be revised carefully.

Author Response

Please see our response and highlighted changes in the attached PDF.

Reviewer 2 Report

Thank you very much for your interesting paper. While reading it, some questions and comments arose which are marked accordingly in the attached pdf file. Please take them into account when revising your manuscript.

In some parts of the text, the wording is somewhat unclear or complicated. The language could be more concise. Some examples can be found in the attached file.

Author Response

(The authors gave the same response as above.)

Round 2

Reviewer 2 Report

Thank you for considering the comments. I think the paper has benefited from them and can now be published in this form.